# Quantum mechanics and the covariance of physical laws in quantum reference frames

Flaminia Giacomini[1,2], Esteban Castro-Ruiz[1,2] & Časlav Brukner[1,2]

In physics, every observation is made with respect to a frame of reference. Although reference frames are usually not considered as degrees of freedom, in all practical situations it is a physical system which constitutes a reference frame. Can a quantum system be considered as a reference frame and, if so, which description would it give of the world? Here, we introduce a general method to quantise reference frame transformations, which generalises the usual reference frame transformation to a "superposition of coordinate transformations". We describe states, measurement, and dynamical evolution in different quantum reference frames, without appealing to an external, absolute reference frame, and find that entanglement and superposition are frame-dependent features. The transformation also leads to a generalisation of the notion of covariance of dynamical physical laws, to an extension of the weak equivalence principle, and to the possibility of defining the rest frame of a quantum system.

[1] Vienna Center for Quantum Science and Technology (VCQ), Faculty of Physics, University of Vienna, Boltzmanngasse 5, A-1090 Vienna, Austria. [2] Institute of Quantum Optics and Quantum Information (IQOQI), Austrian Academy of Sciences, Boltzmanngasse 3, A-1090 Vienna, Austria. Correspondence and requests for materials should be addressed to F.G. (email: flaminia.giacomini@univie.ac.at)

The state of a physical system has no absolute meaning, but is only defined relative to the observer's reference frame in the laboratory. The same system may be associated to different states in different reference frames, which are normally related via some reference frame transformation. From a physical point of view, a frame of reference is an abstraction of an idealised physical system: for example, an ideal rigid body can serve as a reference frame to define relative spatial distances and orientations of other objects. In classical physics, a coordinate transformation is used to transform the description of the system under consideration between two different reference frames. These transformations include, for example, spatial rotations and translations in space and time or constant relative motion of the frames (e.g., Galilean tranformations). In general, the dynamical physical laws are invariant under some group of transformations. For instance, the laws of non-relativistic physics are invariant under Galilean transformations.

In every physical laboratory situation, the reference frame is realised through a physical system. As any physical system, it ultimately behaves according to the laws of quantum mechanics. Therefore, one might see the standard treatment of reference-frame transformations as an approximation to a more fundamental set of transformations. Specifically, one should take into account the possibility that one laboratory, from the perspective of another laboratory, might appear in a superposition or even become entangled with the system. Hence, the relationship between the two laboratories becomes more than a simple coordinate transformation between classical reference frames; it becomes a fundamentally quantum relationship. We may then speak about transformations between 'quantum reference frames' (QRFs). For example, we can imagine that the laboratory and the instruments of one observer are fixed to a platform that is in a superposition of position states with respect to the laboratory of a second observer, as illustrated in Fig. 1. Can we meaningfully define transformations between such QRFs? Which transformations relate quantum states of systems defined with respect to one frame of reference to those defined with respect to a second frame of reference? What are the dynamical physical laws that are invariant under such 'quantum transformations'?

QRFs have been extensively discussed in the literature[1–22]. Previous works on QRFs, and a comparison with our approach, is discussed in detail in Methods (Comparison of previous approaches to quantum reference frames).

In this work, we find (unitary) transformations that relate states, dynamical evolution and measurements from the point of view of different QRFs. This is achieved by changing perspective

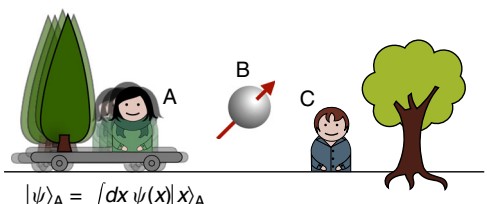

$$|\psi\rangle_A = \int dx\, \psi(x)|x\rangle_A$$

**Fig. 1** Illustration of the notion of quantum reference frames. Two quantum reference frames A and C and quantum system B. The reference frames A and C are pictorially represented as two laboratories equipped with their own instruments. In a realistic situation, however, the system A could be an atom, B a photon and C a laboratory (or another atom). The reference frame associated to A is in a superposition of positions as observed from the laboratory C (the superposition is illustrated by the fuzziness of laboratory A). Given the quantum states of A and B relative to the reference frame of C, what are the states of B and C as defined with respect to the reference frame of A?

via a 'generalised coordinate transformation' from an initial QRF to a final QRF, which does not only involve the observed system, but also the degrees of freedom of the quantum system considered as a reference frame. The resulting transformation takes the states of all systems external to the initial QRF as input, and outputs the states of all systems external to the final QRF. We find that a quantum state and its features—such as superposition and entanglement—are only defined relative to the chosen reference frame, in the spirit of the relational description of physics[16–19,23,24]. For example, a quantum system which is in a well-localised state of an observable for a certain observer may, for another observer, be in a superposition of two or more states or even entangled with the first observer. As the transformations between different QRFs are unitary, the observed probabilities (i.e. relative number of counts) are invariant. However, the measured systems and observables are different in different QRFs and we find the transformation that maps the measured observables and systems in one QRF with those in the other QRF. Turning to the dynamics, we propose an extension of the notion of covariance of the physical laws to include genuine quantum transformations, where one frame of reference is in a superposition of different relative positions, momenta or velocities with respect to another frame of reference. We find Hamiltonians that are symmetric under such "superpositions of Galilean translations" and "superpositions of Galilean boosts". Finally, we find that the weak equivalence principle can be extended to QRFs: The effects as observed in a "superposition of uniform gravitational fields" are indistinguishable from those in a frame in a "superposition of accelerations" in flat space-time. In all these transformations the quantum system considered as a reference frame acts as a control for the transformation on the observed system.

## Results

**Transformations between quantum reference frames**. When reference frames are considered as abstract entities, the reference frame transformation consists in a coordinate transformation, where the new coordinates $x'$ in which the system under consideration is expressed are functions of the old cordinates $x$ and time $t$, i.e. $x' = x'(x, t)$. In the transformation, the relation between the old and new reference frame, such as the relative position or velocity, enters as a parameter. A special case of these transformations, discussed in detail in Supplementary Note 1, are the extended Galilean transformations, introduced in ref.[25]. These are transformations of the type $x' = x - X(t)$, and contain as particular cases spatial translations, Galilean boosts and transformations to a uniformly accelerated reference frame. In quantum theory, all these transformations can be represented via their unitary action on the quantum state of the system $|\psi'\rangle = \hat{U}_i|\psi\rangle$, where the index $i$ labels the transformation. In the case of spatial translations, the coordinate transformation $x' = x - X_0$ induces the transformation $\hat{U}_T = e^{\frac{i}{\hbar}X_0\hat{p}}$, where $X_0$ is a fixed parameter, with the physical dimension of a length, describing the displacement of the new reference frame with respect to the old one. When the new reference frame moves with constant and uniform velocity $v$ from the point of view of the initial one (Galilean boost), we have $X(t) = vt$, and the transformation which changes the state to the new frame is $\hat{U}_b = e^{\frac{i}{\hbar}v\hat{G}}$. Here, $\hat{G} = \hat{p}t - m\hat{x}$ is the generator of the Galilean boost, with $m$ being the mass of the boosted particle. Finally, if the new reference frame is constantly and uniformly accelerated with acceleration $a$ from the point of view of the initial one, $X(t) = \frac{1}{2}at^2$ and the operator to be applied on the state is $\hat{U}_a = e^{-\frac{i}{\hbar}m\dot{X}(t)\hat{x}}e^{\frac{i}{\hbar}X(t)\hat{p}}e^{-\frac{im}{\hbar 2}\int_0^t ds\dot{X}^2(s)}$.

The usual coordinate transformations which describe the change between reference frames rely on the assumption that reference frames are abstract, ideal entities, to which we do not

assign a physical state. In a real experimental situation this idealisation may not be accurate, since real physical systems standardly serve as reference frames, and therefore the assumptions usually made may be untenable. An instance of the differences occurring when a physical system is considered as a reference frame is presented in ref. [26], where a vibrating wire serves as a quantum non-inertial reference frame. The wire, placed in an interferometer traversed by an atom, induces a phase shift on the atom, leading to a loss of interference.

We next give the basic elements of our formalism, where a description of a set of physical systems is given relative to another set of physical systems (the latter set serving as a QRF), within the framework of either classical or quantum theory. We find general transformations between the descriptions that different QRFs provide for their respective 'rests of the world'. We will see that the notion of 'jumping' to a QRF becomes ill-defined: not all the variables can be cast in relational terms, and a choice has to be made as to which degrees of freedom are relevant to the situation studied. All calculations are done in one dimension to keep the notation simple. An extension to three dimensions is straightforward.

We consider three quantum systems, as illustrated in Fig. 1: C is the initial reference frame, A is the new reference frame to whose perspective we want to change, and B is, in general, a composite system to which the transformation from C's to A's reference frame will be applied. Our approach is operational in that primitive laboratory operations—preparations, transformations and measurements—have fundamental status. This emphasis on the operational approach enables the theory to be specified purely in terms of notions that have immediate physical meaning. Note, however, that the approach does not entail the necessity of having macroscopic superpositions. In a realistic situation, for example, system A could be a particle with external degrees of freedom in superposition with respect to laboratory C, which serve to define the new set of relative coordinates, and with internal degrees of freedom used as a 'detector' in reference frame A. We assume that a dynamical description relative to a reference frame does not involve the frame itself, but only the systems external to it. An explanation of this is that, for instance, the position and momentum of the reference frame are not dynamical variables when considered from the reference frame itself (this can also be related to the so-called self-reference problem[27,28]). Therefore, the reference frame is not a degree of freedom in its own description, but external systems to it are. Hence, from the perspective of C's reference frame, A and B are external systems, and from the perspective of A's reference frame so are B and C.

In C's reference frame the systems A and B are described by quantum states in the Hilbert space $\mathcal{H}_A^{(C)} \otimes \mathcal{H}_B^{(C)}$. To change the reference frame we apply a canonical transformation, the most general transformation which preserves the symplectic structure of the phase space. Quantum canonical transformations have been object of study in Refs. [29,30], and are defined as invertible transformations $\hat{C}$ which map the initial operators $(\hat{x}, \hat{p})$, to $\hat{q} = \hat{C}\hat{x}\hat{C}^{-1}$ and $\hat{\pi} = \hat{C}\hat{p}\hat{C}^{-1}$ such that $[\hat{q}, \hat{\pi}] = i\hbar$. The general theory of quantum canonical transformations involves technical issues, for example that not all quantum canonical transformations are isometries[30]. For simplicity, in this work we restrict our consideration only to unitary transformations, which by definition are isometries. Such unitary transformations take the form $\hat{C} : \mathcal{H}_1 \to \mathcal{H}_2$, where $\mathcal{H}_1$, $\mathcal{H}_2$ are the initial and final Hilbert spaces, such that, for all states $\psi$, $\phi \in \mathcal{H}_1$, the scalar product is preserved, i.e. $\langle\phi|\psi\rangle_1 = \langle\hat{C}\phi|\hat{C}\psi\rangle_2$, where $\langle\cdot|\cdot\rangle_i$ is the scalar product on the Hilbert space $\mathcal{H}_i$, $i = 1, 2$. Notice that the functional form of the two scalar products might differ, because the measure of the Hilbert space is allowed to change.

Before giving an example of a transformation to a quantum reference frame, we should stress that the requirement of canonicity leads to important consequences. If we were to approach the transformation naively, we would be tempted to define a transformation to relative coordinates and momenta, where the relative coordinates of a set of $N$ particles of mass $m_i$ ($i = 1,...,N$) relative to particle 0 of mass $m_0$ are $x_i^r = x_i - x_0$ and the relative momenta are $p_i^r = \mu_{i0}\left(\frac{p_i}{m_i} - \frac{p_0}{m_0}\right)$, $\mu_{i0} = \frac{m_i m_0}{m_i + m_0}$ being the reduced mass. If we now compute the Poisson bracket, we find that $\{x_i^r, p_j^r\} \neq 0$ for $i \neq j$, thus violating the canonicity requirement. This argument can also be found in ref. [20,21]. This means that, whenever we perform a transformation to a quantum reference frame, we have to choose the relative variables we are interested in, and then complete the transformation of the conjugated variable by canonicity. Note that this feature does not only arise in quantum mechanics, but in classical physics too. Moreover, in the Lagrange formalism, a transformation to the relative variables is a point transformation, and is therefore automatically canonical when mapped to the phase space.

We now move on to illustrate the idea of a transformation between QRFs through an example. We describe the situation in which the new reference frame A is simply translated with respect to the old one C, but where the quantum state of A, instead of being sharp in position, is in a superposition of positions. In this case it is clear that a mere coordinate transformation of the type discussed previously in this section and in detail in Supplementary Note 1 is no longer adequate, because the position of the new reference frame is not localised, and therefore there is no unique distance between the two reference frames. As a consequence, a transformation which captures the quantum features of A is necessary in order to describe a quantum system B in the new reference frame. We suggest that a natural procedure is to make use of the linearity of quantum mechanics and 'coherently translate' the state of B relative to the position of A, via the operator $e^{\frac{i}{\hbar}\hat{x}_A\hat{p}_B}$, where the indices refer to the two quantum systems A and B. Note that here the position operator of the system A, $\hat{x}_A$, replaces the classical parameter of the usual translation operator.

The full spatial translation to change from C's to A's reference frame consists of a change of relative position coordinates as seen from C (illustrated in Fig. 2a) to the relative position coordinates as seen from A (in Fig. 2b). This can be achieved via the canonical transformation

$$\hat{x}_B \mapsto \hat{q}_B - \hat{q}_C, \tag{1a}$$

$$\hat{x}_A \mapsto -\hat{q}_C, \tag{1b}$$

$$\hat{p}_B \mapsto \hat{\pi}_B, \tag{1c}$$

$$\hat{p}_A \mapsto -(\hat{\pi}_C + \hat{\pi}_B), \tag{1d}$$

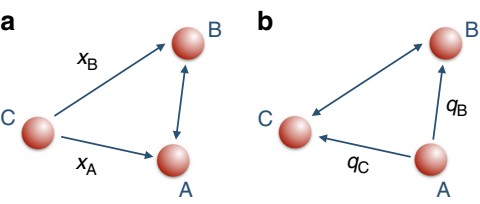

**Fig. 2** Transformation to relative coordinates. **a** Relative position coordinates of A and B from the point of view of C. **b** Relative position coordinates of B and C from the point of view of A. It is immediate to verify that $\hat{x}_B \mapsto \hat{q}_B - \hat{q}_C$ and that $\hat{x}_A \mapsto -\hat{q}_C$

where $\hat{x}_i, \hat{p}_i$, $i = $ A, B, are the positions and momenta relative to C, and $\hat{q}_j, \hat{\pi}_j$, $j = $ B, C, are the positions and momenta relative to A. The transformation (1) can be represented via the canonical transformation $\mathcal{S}_{\mathrm{x}}(\cdot) = \hat{S}_{\mathrm{x}} \cdot \hat{S}_{\mathrm{x}}^{\dagger}$ on the phase space observables of the systems A and B, where $\mathcal{S}_{\mathrm{x}}(\cdot)$ is a superoperator and $\hat{S}_{\mathrm{x}}$ the unitary transformation $\hat{S}_{\mathrm{x}} : \mathcal{H}_{\mathrm{A}}^{(\mathrm{C})} \otimes \mathcal{H}_{\mathrm{B}}^{(\mathrm{C})} \rightarrow \mathcal{H}_{\mathrm{B}}^{(\mathrm{A})} \otimes \mathcal{H}_{\mathrm{C}}^{(\mathrm{A})}$ defined as

$$\hat{S}_{\mathrm{x}} = \hat{\mathcal{P}}_{\mathrm{AC}} e^{\frac{\mathrm{i}}{\hbar}\hat{x}_{\mathrm{A}}\hat{p}_{\mathrm{B}}}. \qquad (2)$$

Here, $\hat{\mathcal{P}}_{\mathrm{AC}} : \mathcal{H}_{\mathrm{A}}^{(\mathrm{C})} \rightarrow \mathcal{H}_{\mathrm{C}}^{(\mathrm{A})}$ is the parity-swap operator acting as $\hat{\mathcal{P}}_{\mathrm{AC}}\psi_{\mathrm{A}}(x) = \psi_{\mathrm{C}}(-x)$ in the coordinate representation of the states. If C assigns the quantum state $\rho_{\mathrm{AB}}^{(\mathrm{C})}$ to the joint system of A and B, the transformed state from A's perspective is $\rho_{\mathrm{BC}}^{(\mathrm{A})} = \hat{S}_{\mathrm{x}}\rho_{\mathrm{AB}}^{(\mathrm{C})}\hat{S}_{\mathrm{x}}^{\dagger}$. Note that our transformation can be applied to both pure and mixed states. This transformation satisfies the transitive property, meaning that changing reference frame from C to B and subsequently from B to A with Eq. (2) has the same effect as changing it from C to A. In particular, this means that changing from C to A and then back to C is equivalent to an identity operation, i.e. $\hat{S}_{\mathrm{x}}^{(\mathrm{C}\rightarrow\mathrm{A})} = \left(\hat{S}_{\mathrm{x}}^{(\mathrm{A}\rightarrow\mathrm{C})}\right)^{\dagger}$. This is shown in Supplementary Note 2. The transformation in Eq. (2) can be seen as a translation of system B controlled by the position of system A followed by the parity-swap operator.

Note that no absolute reference frame ('external' perspective) was needed to establish Eq. (2) (throughout the paper, we interchangeably use the terminology 'absolute', 'abstract', 'external', and 'classical' to refer to such reference frames). This transformation can be obtained by performing a point transformation to relative coordinates in the Lagrangian formalism, from which the relations between momenta can be derived. Alternatively, in the Hamiltonian formalism, the transformation in Eq. (2) can be fixed uniquely (up to a constant in Eq. (2)) by requiring that it is canonical, linear in phase-space observables, and does not mix coordinates and momenta. In this paper, we work with infinite-dimensional Hilbert spaces isomorphic to $L^2(\mathbb{R})$ with standard measure $d\mu(x) = dx$. It may happen that the system constituting the new reference frame, say A, is a composite system from the point of view of C. For instance, A could be composed of different particles (for example, imagine an atom made of protons and neutrons). Our formalism can be applied to this situation by defining the transformation to jump, for instance, to the degrees of freedom of the centre of mass. In this case, the internal degrees of freedom can be transformed like any external system B. In Methods (Application: notion of rest frame of a quantum system) we provide another example of how our formalism can be applied to a composite system with discrete internal degrees of freedom.

The general procedure that we follow to perform the canonical transformation is to choose a basis in which we want to express the relative quantities, and then complete it canonically. Note that any quadrature in the phase space could be considered as the relative variable. Different choices of relative coordinates would induce different transformations between QRFs. In Eq. (1) we have chosen position basis to define the relative coordinates in C and A, but we could have chosen the eigenbasis of, for instance, relative momenta. In this case the transformation is $\hat{S}_{\mathrm{p}} = \hat{\mathcal{P}}_{\mathrm{AC}} e^{-\frac{\mathrm{i}}{\hbar}\hat{p}_{\mathrm{A}}\hat{x}_{\mathrm{B}}}$, and it gives rise to the following canonical transformation: $\hat{p}_{\mathrm{B}} \mapsto \hat{\pi}_{\mathrm{B}} - \hat{\pi}_{\mathrm{C}}$, $\hat{p}_{\mathrm{A}} \mapsto -\hat{\pi}_{\mathrm{C}}$, $\hat{x}_{\mathrm{B}} \mapsto \hat{q}_{\mathrm{B}}$, and $\hat{x}_{\mathrm{A}} \mapsto -(\hat{q}_{\mathrm{C}} + \hat{q}_{\mathrm{B}})$. The possibility of choosing different relative coordinates shows that, when we promote a physical system to a reference frame, the question what the description of the rest of the world is relative to the reference frame is ill-posed unless a choice of relative coordinates is met. An equivalent statement is that, when the reference frame is considered as a physical system,

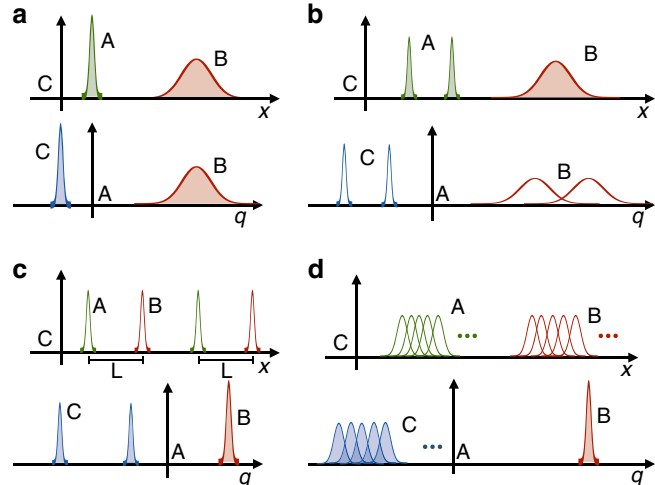

**Fig. 3** Examples of relative states in different quantum reference frames. The relative states are described from the reference frame of C (above in each subfigure) and A (below in each subfigure) in the position basis. Product states are represented as curves whose area is shaded, while entangled states as curves whose area is not shaded. In **a** A's state is well-localised from the point of view of C. In A's reference frame, B has the same state as seen from C, but translated, and C is well-localised. This case corresponds to the translation of a classical reference frame. In **b** A and B are in a product state, and A is in a superposition of two sharp-position states that do not overlap. From A's point of view B and C are entangled, but the relative distance between the states is unchanged. In **c** A and B are entangled and perfectly correlated, i.e. the relative distance between them is always $L$. In A's reference frame B is in a well-defined position and C is in a superposition of positions. Finally, in **d** A and B are entangled in an EPR state from C's point of view, i.e. $|\psi\rangle_{\mathrm{AB}} = \int dx |x\rangle_{\mathrm{A}} |x + X\rangle_{\mathrm{B}}$. Changing to A, B appears in a fixed position, while C is spread over the whole space

there is no unambiguous notion of 'jumping' to a reference frame. Note that this feature arises both in classical and quantum mechanics from the requirement of canonicity of the reference-frame transformation when the reference frames are considered as physical degrees of freedom, and therefore attributed a phase space. The expression "jumping" to a QRF is to be intended, in the rest of the paper, in a loose sense, up to the choice of a specific transformation and basis.

Some examples of transformed states according to the map in Eq. (2) are given in Fig. 3. In particular, we see in Fig. 3a that when the new reference frame A is very sharp in position basis and the initial state in C's reference frame is $|\psi\rangle_{\mathrm{AB}} = |x_0\rangle_{\mathrm{A}}|\psi\rangle_{\mathrm{B}}$, from A's point of view the state of B is translated by $x_0$, and the state of C is also sharp. The state in the new reference frame would then be $|\psi\rangle_{\mathrm{BC}} = \int dq_{\mathrm{B}}\psi(q_{\mathrm{B}} + x_0)|q_{\mathrm{B}}\rangle_{\mathrm{B}}|-x_0\rangle_{\mathrm{C}}$. This corresponds to the translation of a classical reference frame by an amount $x_0$, since transformation $\hat{S}_{\mathrm{x}}$ applied to the well-localised state of A takes the form of the standard translation operator $\hat{S}_{\mathrm{x}}|x_0\rangle_{\mathrm{A}} = \hat{\mathcal{P}}_{\mathrm{AC}} e^{\frac{\mathrm{i}}{\hbar}x_0\hat{p}_{\mathrm{B}}}|x_0\rangle_{\mathrm{A}}$. (Up to the parity-swap operator that specifies the relative position of the two reference frames, which is usually ignored in the standard framework.)

In Fig. 3b we illustrate the case in which the state of A is a superposition of two sharp states, i.e. $|\phi\rangle_{\mathrm{A}} = \frac{1}{\sqrt{2}}\left(|x_1\rangle_{\mathrm{A}} + |x_2\rangle_{\mathrm{A}}\right)$. In general, if C describes the joint state of A and B as a product state $|\phi\rangle_{\mathrm{A}}|\psi\rangle_{\mathrm{B}}$, the state in the reference frame of A is entangled and is obtained as the convolution product of the two, $\hat{S}_{\mathrm{x}}|\phi\rangle_{\mathrm{A}}|\psi\rangle_{\mathrm{B}} = \int dq_{\mathrm{B}} dq_{\mathrm{C}}\phi(-q_{\mathrm{C}})\psi(q_{\mathrm{B}} - q_{\mathrm{C}})|q_{\mathrm{B}}\rangle_{\mathrm{B}}|q_{\mathrm{C}}\rangle_{\mathrm{C}}$. Analogously, if the states of A and B are entangled in the initial reference frame, this property might not hold after changing to the reference frame of A. Examples of this situation are given

in Fig. 3c, d. In particular, in Fig. 3c we consider an entangled state of A and B in position basis, where there is a perfect correlation between A and B, i.e. $|\psi\rangle_{AB} = \frac{1}{\sqrt{2}}\left(|x_1\rangle_A |x_1 + L\rangle_B + |x_2\rangle_A |x_2 + L\rangle_B\right)$. From the point of view of A, the state of B and C is in a product state. In particular, B appears localised at the position $q_B = L$, while the state of C is in the superposition state $\frac{1}{\sqrt{2}}\left(|-x_1\rangle_C + |-x_2\rangle_C\right)$. Similarly, if A and B are entangled in the EPR state $|\psi\rangle_{AB} = \int dx |x\rangle_A |x + X\rangle_B$ as in Fig. 3d, A sees B localised at position $q_B = X$, while C is spread over the whole space.

From the examples considered it is clear that the notions of superposition and entanglement are reference-frame dependent. Specifically, this fact implies that a quantum particle in a spatial superposition from the point of view of a laboratory would in turn attribute to the laboratory a state which is in a spatial superposition. This notion of frame-dependence of the features of a quantum state is different to the one typically found in the literature, where the frame-dependence always appears due to the decoherence of the state of the system as specified in the external reference frame and after tracing out this frame (see, e.g., refs. [4,8,20,21].). (More details on the differences between our approach and the existing literature on QRFs can be found in Methods (Comparison of previous approaches to quantum reference frames))

The dependence of entanglement and superposition on the reference frame is known to appear in relativistic quantum theory due to relativity of simultaneity in different reference frames[31]. Frame-dependent entanglement between momentum and spin degrees of freedom also appears in relativistic quantum information when a state is Lorentz-boosted to a reference frame moving with constant and uniform velocity with respect to the initial one[32]. The boost on the state is represented through a Wigner rotation, which couples the spin and the momentum. Here we show that the effect can arise due to genuine quantum relationships between reference frames even in non-relativistic quantum mechanics.

**Dynamics and symmetries from a quantum reference frame**. We will now derive the Schrödinger equation as seen from a QRF. More specifically, starting from the Hamiltonian in the initial reference frame C, we will derive the dynamical law relative to A. Here, the transformation $\hat{S}$ from C to A is a completely general unitary operator, and may depend explicitly on time. In addition, we assume that the states of A and B with respect to the reference frame C satisfy the Schrödinger equation:

$$i\hbar \frac{d\hat{\rho}_{AB}^{(C)}}{dt} = \left[\hat{H}_{AB}^{(C)}, \hat{\rho}_{AB}^{(C)}\right], \quad (3)$$

where $\hat{\rho}_{AB}^{(C)}$ is the state of A and B (which can be either pure or mixed) and $\hat{H}_{AB}^{(C)}$ their Hamiltonian relative to C.

The state of B and C relative to A is given by $\hat{\rho}_{BC}^{(A)} = \hat{S}\hat{\rho}_{AB}^{(C)}\hat{S}^\dagger$. Differentiating this expression with respect to time and applying the Leibniz rule we obtain

$$i\hbar \frac{d\hat{\rho}_{BC}^{(A)}}{dt} = \left[\hat{H}_{BC}^{(A)}, \hat{\rho}_{BC}^{(A)}\right], \quad (4)$$

where

$$\hat{H}_{BC}^{(A)} = \hat{S}\hat{H}_{BC}^{(A)}\hat{S}^\dagger + i\hbar \frac{d\hat{S}}{dt}\hat{S}^\dagger. \quad (5)$$

At first sight, the Schrödinger equation in Eq. (3) for a general quantum reference frame C might look unjustified. According to our current status of experimental tests, the evolution of a

quantum system has been confirmed as unitary only with respect to an 'abstract' reference frame (in the sense explained in the Introduction). Such an abstract reference frame can be approximated, in our description, as a very massive and classical-like reference frame. Then the mathematical steps from Eqs. (3)–(5) show that, starting from a classical-like reference frame, the evolution can be described unitarily also from any QRF, including those which are not massive, provided that there can exist a transformation like the $\hat{S}$ operator to change the reference frame. Therefore, C can be taken as a QRF also in Eq. (3).

A transformation that leaves the Hamiltonian invariant is called a symmetry transformation. The symmetry of a transformation implies the existence of a conservation of a dynamical observable of the system. In the next sections we will identify symmetry transformations between QRFs. An important difference between classical and quantum RF transformations is that, in the latter case, the Hamitonian as seen from one QRF not only includes the observed system B but also the other QRF. We then define a symmetry transformation as a map that leaves the functional form of the Hamiltonian invariant, i.e. the Hamiltonian of A and B is the same function of operators as the Hamiltonian of C and B,

$$\hat{S}\hat{H}\left(\{m_i, \hat{x}_i, \hat{p}_i\}_{i=A,B}\right)\hat{S}^\dagger + i\hbar \frac{d\hat{S}}{dt}\hat{S}^\dagger = \hat{H}\left(\{m_i, \hat{q}_i, \hat{\pi}_i\}_{i=B,C}\right), \quad (6)$$

where the operators and the mass of A are simply replaced by the ones of C. It can be shown that if condition (6) is satisfied, then the transformation $\hat{S}$ allows to define a map between the dynamical conserved quantities in the reference frame C, $\left\{\hat{C}_i^{(C)}\right\}_{i=1,\ldots,N}$, with $N \in \mathbf{N}$, to the dynamical conserved quantities in the reference frame A, $\left\{\hat{C}_i^{(A)}\right\}_{i=1,\ldots,N}$. In particular, these quantities have the same functional form, but with all the labels A and C swapped. We show this in Supplementary Note 3.

As discussed at the beginning of Results (Transformations between quantum reference frames) and in detail in Supplementary Note 1, when reference frames are treated as abstract entities, the relationship between the old and the new reference frame enters as a function in the transformation $\hat{U}_i = \Pi_n e^{\frac{i}{\hbar} f^n(t)\hat{O}_B^n}$, where $f^n(t)$ depends on the specific transformation between two reference frames, and specifically on the displacement of the two reference frames $X(t)$ and its time-derivatives. The operator $\hat{O}_B$ acts on B's Hilbert space. The product over the index $n$ is due to the fact that, in a general transformation, we might not be able to decompose the transformation into a single product of a function of A and an operator of B. This condition translates to our formalism by promoting the functions $f^n(t)$ to time-dependent operators $\hat{f}_A^n(t)$. This transformation is specified in the time-dependent operators of A, i.e. in the Heisenberg picture. We want to apply the transformation $\hat{S}$ to the states of A and B at time $t$, i.e. to their states in the Schrödinger picture, and to obtain the transformed state of B and C at time $t$. This implies the following structure of general $\hat{S}$ operator:

$$\hat{S} = e^{-\frac{i}{\hbar}\hat{H}_C t}\mathcal{P}_{AC}^{(i)}\Pi_n e^{\frac{i}{\hbar}\hat{f}_A^n(t)\hat{O}_B^n}e^{\frac{i}{\hbar}\hat{H}_A t}, \quad (7)$$

where the prescription to change QRF can be described through the following steps: (a) we first map the state of A to the Heisenberg picture by evolving it back in time with the Hamiltonian $\hat{H}_A$, (b) we then apply the generalisation of the classical transformation using the operators $\hat{f}_A^n(t)$ and (c) we apply the 'generalised parity operator' $\mathcal{P}_{AC}^{(i)}$ to exchange the equations of motion of C and A (e.g. depending on the specific transformation $i$ chosen, the parity operator ensures that the position, velocity or acceleration of A from the point of view of C

| | | Action on system B (action on A is omitted) | Classical RF transformation |
|---|---|---|---|
| Relative coordinates | $\hat{S}_x$ | $\hat{x}_B \mapsto \hat{q}_B - \hat{q}_C$ <br> $\hat{p}_B \mapsto \hat{\pi}_B$ | $\hat{U}_x = e^{\frac{i}{\hbar}X(t)\hat{p}}$ |
| Superposition of translations | $\hat{S}_T$ | $\hat{x}_B \mapsto \hat{q}_B - \hat{q}_C + \frac{\hat{\pi}_C}{m_C}(t-\tau)$ <br> $\hat{p}_B \mapsto \hat{\pi}_B$ | $\hat{U}_T = e^{\frac{i}{\hbar}X_0\hat{p}}$ |
| Superposition of boosts | $\hat{S}_b$ | $\hat{x}_B \mapsto \hat{q}_B - \frac{\hat{\pi}_C}{m_C}t$ <br> $\hat{p}_B \mapsto \hat{\pi}_B - \frac{m_B}{m_C}\hat{\pi}_C$ | $\hat{U}_b = e^{\frac{i}{\hbar}v(t\hat{p}-m\hat{x})}$ |
| Superposition of accelerations | $\hat{S}_{EP}$ | $\hat{x}_B \mapsto \hat{q}_B - \frac{\hat{\pi}_C}{m_C}t + \frac{1}{2m_A}\frac{dV(\hat{q}_C)}{d\hat{q}_C}t^2$ <br> $\hat{p}_B \mapsto \hat{\pi}_B - \frac{m_B}{m_C}\hat{\pi}_C$ | $\hat{U}_a = e^{-\frac{i}{\hbar}mat\hat{x}}e^{\frac{i}{\hbar}\frac{at^2}{2}\hat{p}}e^{-\frac{i}{\hbar}\frac{ma^2t^3}{6}}$ |

**Fig. 4** Table summarising different quantum reference frame (QRF) transformations on system B. (The action on A is omitted) The time-independent transformation $\hat{S}_x$ is the QRF generalisation of a standard reference frame (RF) transformation where the reference frame is moving along $X(t)$, and the relative coordinates describe the distance between systems A and B at time $t$. The three QRF transformations $\hat{S}_T$, $\hat{S}_b$, and $\hat{S}_{EP}$ generalise the extended Galilean transformations to a reference frame which is respectively translated, moving with constant and uniform velocity, and moving with constant and uniform acceleration. In particular, $\hat{S}_T$ transforms the old coordinate $x_B$ into the relative position between system B at time $t$ and system A at time $\tau$, and reduces to $\hat{S}_x$ for $t=\tau$. The transformation $\hat{S}_b$ performs a Lorentz boost on system B, where the velocity is written in terms of dynamical variables of system A. Finally, the transformation $\hat{S}_{EP}$, generalises the transformation to an accelerated reference frame to when system A moves in a superposition of accelerations. Importantly, the extension of the RF transformation to dynamical quantities of quantum systems makes it possible to introduce a generalised notion of symmetric transformation, which is exemplified, in this work, in the case of the $\hat{S}_T$ and $\hat{S}_b$ transformations

are opposite to the same quantities of C from the point of view of A, see below for more details); finally (d) we map the state of C back to the Schrödinger picture via the Hamiltonian $\hat{H}_C$.

Equation (7) is in fact the most general QRF transformation achievable in one dimension when A and B do not interact, and contains as particular cases the (extended) Galilean transformations in one dimension. Via this transformation, it is possible to identify a general method to quantise a reference frame transformation. Firstly, one should identify the type of transformation (e.g., translation, boost, accelerated reference frame, etc.). Secondly, by solving the equations of motion of the system constituting the new reference frame, one should quantise the dynamical variables of the classical reference frame appearing in the transformation by using the phase space operators of the new QRF. This constitutes the central part of Eq. (7). Finally, one should add the two propagators with Hamiltonians $\hat{H}_A$ and $\hat{H}_C$ and choose a generalised parity-swap operator in such a way that the solutions of the equations of motion of system C from the point of view of A are of opposite sign to those of the equations of motion of A from the point of view of C, i.e., they are equal up to a minus sign.

We next exemplify this procedure through the generalisation of the extended Galilean transformations to QRFs, which is discussed in detail in Methods and summarised in Fig. 4. In particular, the Galilean translations can be generalised by considering the relative coordinates as defined relative to the quantum state of a system A at some time $\tau$ (see Methods (Translations between quantum reference frames) for more details). Following the general scheme introduced in Eq. (7), this transformation can be written as

$$\hat{S}_T = \exp\left(-\frac{i}{\hbar}\frac{\hat{\pi}_C^2}{2m_C}(t-\tau)\right)\mathcal{P}_{AC}^{(x)}\exp\left(\frac{i}{\hbar}\hat{x}_A\hat{p}_B\right)\exp\left(\frac{i}{\hbar}\frac{\hat{p}_A^2}{2m_A}(t-\tau)\right). \quad (8)$$

In this case, the QRF is provided by the quantum state of system A at some fixed time $\tau$. The transformation $\hat{S}_T$ maps the

position $x_B$ of particle B at time $t$ in C into the relative position of B at time $t$ and the position of C at time $\tau$ (see Fig. 4 and Methods). In addition, this transformation is a symmetry, in the extended sense of QRFs, for the free particle Hamiltonian, because it maps $\hat{H}_{AB}^{(C)} = \frac{\hat{p}_A^2}{2m_A} + \frac{\hat{p}_B^2}{2m_B}$ into $\hat{H}_{BC}^{(A)} = \frac{\hat{\pi}_B^2}{2m_B} + \frac{\hat{\pi}_C^2}{2m_C}$.

The second case we consider is the generalisation of the Galilean boosts to the "superposition of boosts," corresponding to when particle A moves in a superposition of momenta (velocities) from the point of view of C. This case is discussed in detail in Methods (Boosts between quantum reference frames). In this case, the QRF transformation is

$$\hat{S}_b = \exp\left(-\frac{i}{\hbar}\frac{\hat{\pi}_C^2}{2m_C}t\right)\mathcal{P}_{AC}^{(v)}\exp\left(\frac{i}{\hbar}\frac{\hat{p}_A}{m_A}\hat{G}_B\right)\exp\left(\frac{i}{\hbar}\frac{\hat{p}_A^2}{2m_A}t\right), \quad (9)$$

and has the physical meaning of "jumping" to the rest frame of a quantum system described by a free-particle Hamiltonian $\hat{H}_{AB}^{(C)} = \frac{\hat{p}_A^2}{2m_A} + \frac{\hat{p}_B^2}{2m_B}$. This transformation is also a symmetry for the free-particle Hamiltonian, which is transformed into $\hat{H}_{BC}^{(A)} = \frac{\hat{\pi}_B^2}{2m_B} + \frac{\hat{\pi}_C^2}{2m_C}$.

Finally, we generalise the transformation to a constantly accelerated reference frame to the QRF transformation $\hat{S}_{EP}$ (see Methods (The weak equivalence principle in quantum reference frames)), describing the change to a QRF moving in a superposition of accelerations. This superposition is achieved by choosing as initial Hamiltonian $\hat{H}_{AB}^{(C)} = \hat{H}_A + \hat{H}_B = \frac{\hat{p}_A^2}{2m_A} + \frac{\hat{p}_B^2}{2m_B} + V(\hat{x}_A)$, where $V(\hat{x}_A)$ is piecewise linear, and the system A evolves in a superposition of two amplitudes, each of which is localised in a region corresponding to a single gradient of the potential. The QRF transformation is expressed as

$$\hat{S}_{EP} = e^{-\frac{i}{\hbar}\left(\frac{\hat{\pi}_C^2}{2m_C}+\frac{m_C}{m_A}V(-\hat{q}_C)\right)t}\mathcal{P}_{AC}^{(v)}\hat{Q}_t e^{\frac{i}{\hbar}\left(\frac{\hat{p}_A^2}{2m_A}+V(\hat{x}_A)\right)t}, \quad (10)$$

where

$$\hat{Q}_t = e^{-\frac{i}{\hbar}\frac{m_B}{m_A}\left(\hat{p}_A - \frac{dV(\hat{x}_A)}{dx_A}t\right)\hat{x}_B} e^{\frac{i}{\hbar}\left(\hat{p}_A - \frac{1}{2}\frac{dV(\hat{x}_A)}{dx_A}t\right)\frac{\hat{p}_B}{m_A}t} e^{-\frac{i}{\hbar}\frac{m_B}{2m_A^2}\int_0^t ds\left(\hat{p}_A - \frac{dV(\hat{x}_A)}{dx_A}s\right)^2}$$

(11)

is the straightforward extension of the usual transformation to an accelerated reference frame. The new Hamiltonian from the point of view of A is

$$\hat{H}_{BC}^{(A)} = \frac{\hat{\pi}_B^2}{2m_B} + \frac{\hat{\pi}_C^2}{2m_C} + \frac{m_C}{m_A}V(-\hat{q}_C) - \frac{m_B}{m_A}\frac{dV}{d\hat{x}_A}\Big|_{-\hat{q}_C}\hat{q}_B, \qquad (12)$$

where a non-inertial term appears, and particle B moves in a gravitational potential with gravitational acceleration depending on the acceleration of particle C. Once the transformation of the quantum state is taken into account, it is possible to see (see Methods for details) that the system B evolves as if it were in a superposition of uniform gravitational fields. The weak equivalence principle states that physical laws as seen from a reference frame moving with a constant and uniform acceleration are indistinguishable from those as seen in an uniform gravitational field. Our result extends the principle to the equivalence between the physical laws as seen from a QRF in a superposition of constant and uniform accelerations and those as seen in a superposition of uniform gravitational fields.

This completes our discussion of how the extended Galilean transformations can be generalised to QRFs. In all the examples provided, we considered situations in which A and B initially do not interact. The most general case, in which systems A and B evolve in a general, interacting potential, will be object of future investigation.

This method to quantise a reference frame transformation allows us to define the transformation to the rest frame of a quantum system, e.g., a system moving in a superposition of velocities from the point of view of the laboratory. This has important applications in the study of the internal degrees of freedom of quantum systems. In Methods (Application: notion of rest frame of a quantum system) we provide an example of such a situation. Finally, in Methods (Measurements as seen from a quantum reference frame) we show how the description of a measurement procedure changes with the change of a QRF.

## Discussion

In this work we introduced an operational formalism to apply quantum mechanics from the point of view of a reference frame attached to a quantum particle, which we call quantum reference frame. This reference frame has its own degrees of freedom, which can be in quantum superposition or entangled and evolve in time according to their own Hamiltonian with respect to the laboratory frame of reference. We adopt a relational view, according to which any reference frame is described as a quantum degree of freedom relatively to another reference frame: hence, the laboratory frame of reference is a quantum system relative to the quantum reference frame of a particle, much like the particle is a quantum system relative to the laboratory frame. This allows us to avoid assuming the existence of an 'external' perspective of an absolute reference frame.

We find transformations between quantum reference frames, and show how the state, the dynamics, and the measurement change under these transformations. We show that the notion of entanglement and superposition are observer-dependent features, and we write the Schrödinger equation in quantum reference frames. Furthermore, we introduce a generalised notion of covariance of physical laws for quantum reference frames. We apply

our formalism to the situations in which the reference frames are related via 'superposition of translations' and 'superposition of Galilean boosts', and formulate an extension of the weak equivalence principle for such quantum reference frames.

This work has been carried out within Galilean relativity, however the framework is general and can be applied in a special-relativistic or in a general-relativistic context. This would lead to interesting insights as to, for instance, the flow of proper time when there is no classical worldline describing the motion of the system serving as reference frame. More specifically, our formalism could be able to describe situations, such as those studied in refs. [33–35], in which clocks—quantum systems with internal degrees of freedom—move in superpositions of classical worldlines in the gravitational field. As a result, the clock's internal and external degrees of freedom get entangled, because the clock's proper time depends on the worldline taken in the superposition. In these situations, proper time is measured by the clock in its rest frame, but currently no complete formalism is known which would allow to transform to the rest frame of a clock that is in superposition of positions or momenta with repect to the laboratory frame. Already in the present work we provide a solution to this problem in the low-velocity limit to explain the Doppler-shift induced transitions for atoms in superpositions of momenta (see Methods (Application: notion of rest frame of a quantum system). We move to the rest frame of the atom, compute the transition probabilities for the incoming light frequencies in this frame, and then move back to the laboratory frame.

An alternative future direction of our work concerns the application to future experiments, in particular those able to test relative variables, such as the techniques in refs. [36–39], and those involving 'macroscopic' systems (e.g. nanomechanical oscillators), which could play the role of 'large' quantum reference frames, similarly to the situation considered in ref. [26]. Experiments with these systems could shed light on some conceptual issues of quantum gravity at low energies, such as those related to quantum fluctuations of the spacetime or superposition of large masses.

It would also be interesting to investigate whether allowing observers to be in a superposition or entangled with other systems could lead to scenarios with indefinite causal structures, such as those in ref. [40], where a global time-order cannot, at least in general, be imposed, but the observers are in well-defined positions. Our formalism for quantum reference frames can be seen as a dual picture to this work: while a global time order can still be found, at least in the Galilean-relativistic case, the observers are not localised.

## Methods

**Translation between quantum reference frames**. We first consider the case in which we change to the reference frame described by the position of the quantum system A at a particular instant of time $\tau$, when the initial Hamiltonian for A and B is $\hat{H}_{AB}^{(C)} = \frac{\hat{p}_A^2}{2m_A} + \frac{\hat{p}_B^2}{2m_B}$. In this case, the operator $\hat{X}_A(t)$ generalises the function $X(t) = X_0$, with $X_0$ being a constant, and takes the form $\hat{X}_A(t) = e^{\frac{i}{\hbar}\frac{\hat{p}_A^2}{2m_A}\tau}\hat{x}_A e^{-\frac{i}{\hbar}\frac{\hat{p}_A^2}{2m_A}\tau}$, and the full operator $\hat{S}$ is

$$\hat{S}_T = \exp\left(-\frac{i}{\hbar}\frac{\hat{\pi}_C^2}{2m_C}(t-\tau)\right)\hat{\mathcal{P}}_{AC}^{(x)}\exp\left(\frac{i}{\hbar}\hat{x}_A\hat{p}_B\right)\exp\left(\frac{i}{\hbar}\frac{\hat{p}_A^2}{2m_A}(t-\tau)\right), \qquad (13)$$

where $\hat{\mathcal{P}}_{AC}^{(x)} = \hat{\mathcal{P}}_{AC}$ in Eq. (2) and we have introduced the term $\exp\left(-\frac{i}{\hbar}\frac{\hat{\pi}_C^2}{2m_C}(t-\tau)\right)$ to ensure that the position of the system A at time $\tau$ tranforms into the symmetric position of the system C, i.e. $\hat{S}_T\left(\hat{x}_A - \frac{\hat{p}_A}{m_A}(t-\tau)\right)\hat{S}_T^\dagger = -\left(\hat{q}_C - \frac{\hat{\pi}_C}{m_C}(t-\tau)\right)$. Notice that for $t = \tau$ the operator $\hat{S}_T$ in Eq. (13) is precisely the operator $\hat{S}_x$ in Eq. (2). Therefore, we can interpret $\hat{S}_x$ as the operator which performs the translation to a quantum reference frame when the dynamics is "frozen" at time $\tau$. The

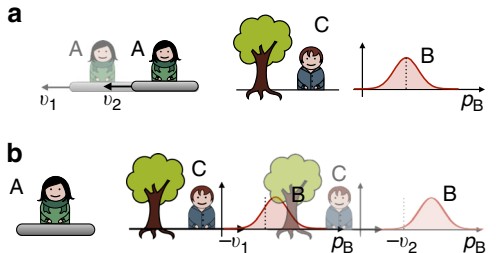

**Fig. 5** Schematic illustration of the descriptions in two quantum reference frames that are boosted with respect to each other: **a** the state of A and B as described from C, and **b** the state of B and C as described from A. In **a**, the state of A and B is a product state, and the state of A is in a superposition of the two velocities $v_1$ and $v_2$. By applying a 'superposition of boosts' by the velocity of A, we find that, as seen from A, the state of B and C is entangled. In particular, the entanglement is such that if C moves with velocity $-v_i$, $i = 1, 2$, from A's point of view, B is boosted by $-v_i$

transformation implemented by $\hat{S}_T$ is

$$\hat{S}_T \hat{x}_A \hat{S}_T^\dagger = -\hat{q}_C + \frac{\hat{\pi}_C}{m_C}(t-\tau) - \frac{\hat{\pi}_B + \hat{\pi}_C}{m_A}(t-\tau); \qquad \hat{S}_T \hat{p}_A \hat{S}_T^\dagger = -(\hat{\pi}_B + \hat{\pi}_C); \tag{14}$$

$$\hat{S}_T \hat{x}_B \hat{S}_T^\dagger = \hat{q}_B - \hat{q}_C + \frac{\hat{\pi}_C}{m_C}(t-\tau); \qquad \hat{S}_T \hat{p}_B \hat{S}_T^\dagger = \hat{\pi}_B. \tag{15}$$

Equation (15) implies that the position at time $t$ of system B from the point of view of C is mapped into the relative position between system B and the position of A at time $\tau$, while the momentum of B remains unchanged. In addition, this transformation is a symmetry of the free particle according to the definition given in Eq. (6), because the Hamiltonian $\hat{H}_{AB}^{(C)}$ is mapped through Eq. (5) to $\hat{H}_{BC}^{(A)} = \frac{\hat{\pi}_B^2}{2m_B} + \frac{\hat{\pi}_C^2}{2m_C}$. Therefore, the transformation $\hat{S}_T$ in Eq. (13) constitutes a generalisation of the Galilean translations to quantum reference frames. The simplest example of dynamical conserved quantities, in this case, are the two momenta $\hat{C}_1^{(C)} = \hat{p}_A$ and $\hat{C}_2^{(C)} = \hat{p}_B$. It is immediate from Eqs. (14) and (15) to see that the choice $\hat{C}_1^{(A)} = \hat{S}_T \hat{C}_2^{(C)} \hat{S}_T^\dagger = \hat{\pi}_B$ and $\hat{C}_2^{(A)} = -\hat{S}_T \hat{C}_1^{(C)} \hat{S}_T^\dagger - \hat{S}_T \hat{C}_2^{(C)} \hat{S}_T^\dagger = \hat{\pi}_C$ leads to the corresponding conserved quantities in the reference frame A. A similar procedure holds when we consider the extended set of conserved quantities composed of translations $\hat{p}_i$ and Galilean boosts $\hat{G}_i = \hat{p}_i t - m_i \hat{x}_i$, $i = A, B$. Notice that this construction of the $\hat{S}_T$ operator satisfies the transitive property, meaning that changing the reference frame from C to A directly has the same effect as changing the reference frame first from C to B and then from B to A, i.e. $\hat{S}_T^{(C \to A)} = \hat{S}_T^{(B \to A)} \hat{S}_T^{(C \to B)}$.

**Boosts between quantum reference frames**. The second example we consider is the change to a reference frame moving with the velocity of a quantum system A, which is described as a free-particle from the point of view of the initial observer C. The total Hamiltonian for both systems A and B from C's point of view is $\hat{H}_{AB}^{(C)} = \frac{\hat{p}_A^2}{2m_A} + \frac{\hat{p}_B^2}{2m_B}$. This section generalises the usual Galilean boost $\hat{U}_b = e^{\frac{i}{\hbar} v \hat{G}_B}$, with $\hat{G}_B = \hat{p}_B t - m_B \hat{x}_B$ being the generator of the boost on system B, introduced in Results (Transformations between quantum reference frames), to situations in which the velocity of the reference frame is distributed according to its quantum state. With reference to Eq. (7), in this case we have $\frac{d\hat{x}_A}{dt} = \frac{\hat{p}_A}{m_A}$ generalising the parameter $v$ in $\hat{U}_b$. The complete transformation, which we call $\hat{S}_b$, is

$$\hat{S}_b = \exp\left(-\frac{i}{\hbar} \frac{\hat{\pi}_C^2}{2m_C} t\right) \mathcal{P}_{AC}^{(v)} \exp\left(\frac{i}{\hbar} \frac{\hat{p}_A}{m_A} \hat{G}_B\right) \exp\left(\frac{i}{\hbar} \frac{\hat{p}_A^2}{2m_A} t\right), \tag{16}$$

where the 'generalised parity operator' $\mathcal{P}_{AC}^{(v)} = \mathcal{P}_{AC} \exp\left(\frac{i}{\hbar} \log\sqrt{\frac{m_C}{m_A}}(\hat{x}_A \hat{p}_A + \hat{p}_A \hat{x}_A)\right)$ maps the velocity of A to the opposite of the velocity of C via the standard parity-swap operator $\mathcal{P}_{AC}$ and an operator scaling coordinates and momenta. Specifically, $\mathcal{P}_{AC}^{(v)} \hat{x}_A \left(\mathcal{P}_{AC}^{(v)}\right)^\dagger = -\frac{m_C}{m_A} \hat{q}_C$, $\mathcal{P}_{AC}^{(v)} \hat{p}_A \left(\mathcal{P}_{AC}^{(v)}\right)^\dagger = -\frac{m_A}{m_C} \hat{\pi}_C$. This choice of $\hat{S}_b$ ensures that the velocity of A in the reference frame of C is opposite to the velocity of C in the reference frame of A, i.e. $\hat{S}_b \frac{\hat{p}_A}{m_A} \hat{S}_b^\dagger = -\frac{\hat{\pi}_C}{m_C}$. The coordinates and

momenta transform as

$$\hat{S}_b \hat{x}_A \hat{S}_b^\dagger = -\frac{m_C \hat{q}_C + m_B \hat{q}_B}{m_A} + \frac{\hat{\pi}_C + \hat{\pi}_B}{m_A} t - \frac{\hat{\pi}_C}{m_C} t, \hat{S}_b \hat{p}_A \hat{S}_b^\dagger = -\frac{m_A}{m_C} \hat{\pi}_C, \tag{17}$$

$$\hat{S}_b \hat{x}_B \hat{S}_b^\dagger = \hat{q}_B - \frac{\hat{\pi}_C}{m_C} t, \qquad \hat{S}_b \hat{p}_B \hat{S}_b^\dagger = \hat{\pi}_B - \frac{m_B}{m_C} \hat{\pi}_C. \tag{18}$$

This transformation, similarly to the transformation $\hat{S}_T$ discussed previously, is also a symmetry of the free-particle Hamiltonian, because it maps the initial Hamiltonian $\hat{H}_{AB}^{(C)} = \frac{\hat{p}_A^2}{2m_A} + \frac{\hat{p}_B^2}{2m_B}$ to the Hamiltonian in the new reference frame $\hat{H}_{BC}^{(A)} = \frac{\hat{\pi}_B^2}{2m_B} + \frac{\hat{\pi}_C^2}{2m_C}$ through Eq. (5). Hence, this constitutes a Galilean boost transformation for quantum reference frames, which allows the system defining the reference frame to be in a superposition of velocities.

To illustrate this point, we consider the situation depicted in Fig. 5. We consider a state $|\Psi_t\rangle_{AB} = e^{-\frac{i}{\hbar} \hat{H}_{AB}^{(C)} t} |\phi_0\rangle_A |\psi_0\rangle_B$, where the initial state $|\phi_0\rangle_A = \int dp_A \phi_0(p_A) |p_A\rangle_A$ of system A is in a superposition of momenta with respect to the initial reference frame C. We now change perspective to the reference frame A. No simple coordinate transformation of reference frames could capture this change. Our method gives, as a result of the transformation $\hat{S}_b$, the entangled state of B and C

$$\hat{S}_b |\Psi_t\rangle_{AB} = \int d\pi_C d\pi_B \pi_C e^{-\frac{i}{\hbar}\left(\frac{\pi_C^2}{2m_C} + \frac{\pi_B^2}{2m_B}\right)t} \phi_0\left(-\frac{m_A}{m_C}\pi_C\right) \psi_0(\pi_B) e^{-\frac{i}{\hbar}\frac{\pi_C}{m_C}\hat{G}_B} |\pi_C\rangle_C |\pi_B\rangle_B.$$

The state of B is boosted by the velocity of A (which corresponds to the opposite of the velocity of C, given Eq. (18)) for each momentum in the superposition state of A, while the system C evolves as a free particle with opposite velocity to A.

In the special case of a free particle B in the general state $|\psi_0\rangle_B$ and the reference frame A having a state with a well-defined momentum (velocity) $|\phi_0\rangle_A = |p_A\rangle_A$, the transformed state $\hat{S}_b |\Psi_t\rangle_{AB} = e^{-\frac{i}{\hbar}\frac{\pi_C^2}{2m_C}t} |\pi_C\rangle_C e^{-\frac{i}{\hbar}\frac{\pi_C}{m_C}\hat{G}_B} |\psi_t\rangle_B$, where $|\psi_t\rangle_B$ is the time evolved state and $\pi_C = -\frac{m_C}{m_A} p_A$, reduces to the standard boost transformation $\hat{U}_b$ in the usual description of reference frames, with the difference that here C is a degree of freedom and hence evolved in time.

With a similar reasoning to the one presented in the previous section, it is possible to show that the set of the conserved quantities $\hat{p}_A$, $\hat{p}_B$, $\hat{G}_A$, $\hat{G}_B$ in the reference frame C is mapped to the set of the conserved quantities in the reference frame A $\hat{\pi}_B$, $\hat{\pi}_C$, $\hat{G}_B$, $\hat{G}_C$. Analogously to the generalised translations in the previous Subsection, this choice of the operator $\hat{S}_b$ also satisfies the transitive property, i.e. $\hat{S}_b^{(C \to A)} = \hat{S}_b^{(B \to A)} \hat{S}_b^{(C \to B)}$.

Notice that a time-independent version of the transformation $\hat{S}_b$, mapping to instantaneous relative velocities, would not preserve the invariance of the Hamiltonian. This example is discussed in Supplementary Note 4.

**The weak equivalence principle in quantum reference frames**. In this section we generalise the weak equivalence principle to quantum reference frames. By this, we mean that the physical effects as seen from a reference frame moving in a superposition of uniform gravitational fields are indistinguishable from those as seen from a system in superposition of accelerations. To achieve a superposition of accelerations, let us consider the situation depicted in Fig. 6, in which two particles A and B evolve in time according to the Hamiltonian

$$\hat{H}_{AB}^{(C)} = \hat{H}_A + \hat{H}_B = \frac{\hat{p}_A^2}{2m_A} + \frac{\hat{p}_B^2}{2m_B} + V(\hat{x}_A) \tag{19}$$

in the reference frame of an observer C.

For the purpose of further analysis we will now consider the potential $V(\hat{x}_A)$ to be piecewise linear and particle A to evolve in time $t$ as a superposition of wave amplitudes, each localised in an interval that corresponds to a constant yet different potential gradient. For concreteness consider the superposition of two such amplitudes, $|\psi_0(t)\rangle_A = \frac{1}{\sqrt{2}}\left(|\psi_1(t)\rangle_A + |\psi_2(t)\rangle_A\right)$ (see Fig. 7). The state then is in a superposition of accelerations, i.e. the 'acceleration operator' applied on the state gives:

$$-\frac{1}{m_A}\frac{dV(\hat{x}_A)}{d\hat{x}_A}|\psi_0(t)\rangle_A \approx \frac{1}{\sqrt{2}}\left(a_1 |\psi_1(t)\rangle_A + a_2 |\psi_2(t)\rangle_A\right), \tag{20}$$

where $a_1 = -\frac{1}{m_A}\frac{dV(\hat{x}_A)}{d\hat{x}_A}\big|_{x_1(t)}$ and $a_2 = -\frac{1}{m_A}\frac{dV(\hat{x}_A)}{d\hat{x}_A}\big|_{x_2(t)}$, where $x_1(t)$ and $x_2(t)$ are the mean values of position operator for the individual localised amplitudes. Notice that the scalar accelerations $a_1$ and $a_2$ as well as the scalar positions $x_1(t)$ and $x_2(t)$ should be understood as multiplied by the identity operator.

In order to find the generalised version of the operator $\hat{U}_a$, discussed in Results (Transformations between quantum reference frames) and in detail in Supplementary Note 1, and get an expression analogous to the one in Eq. (7), we need the time derivative of the position operator $\hat{x}_A$ at time $t$. To calculate the evolved position operator, we write an explicit expression for

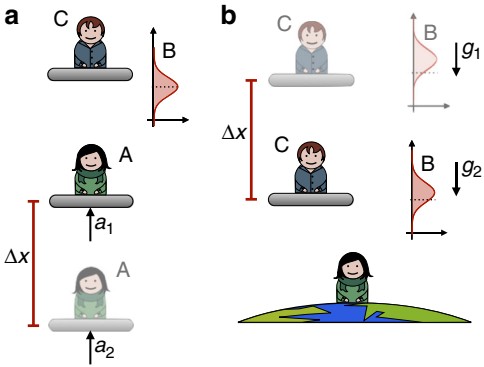

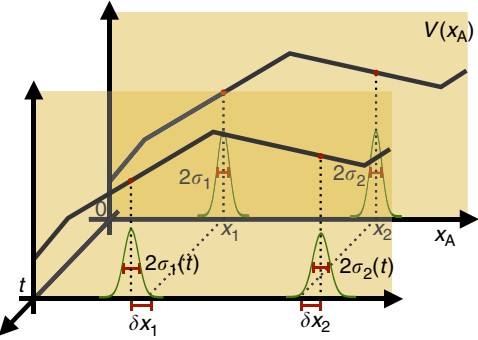

**Fig. 6** Generalisation of the weak equivalence principle for quantum reference frames. The quantum system A is initially in a superposition of two localised wave amplitudes in a piecewise linear potential $V(\hat{x}_A)$ from the point of view of another system C. The individual wave amplitudes are localised in spatial intervals corresponding to two different potential gradients. The system B evolves instead as a free particle. If we consider the motion for sufficiently short times, such that the two amplitudes remain localised within the corresponding intervals, the system A evolves as if it were in a superposition of the accelerations $a_1$ and $a_2$. We can then change perspective to the accelerated reference frame of A, by applying a transformation corresponding to a 'superposition of accelerations', and describe how the quantum system A sees the quantum systems B and C and their evolution. After the transformation, the system B evolves as if it was moving in a superposition of linear gravitational potentials, where the gravitational accelerations are such that $\vec{g}_i = -\vec{a}_i$, where $i = 1, 2$. This means that the effects of a superposition of accelerations are indistinguishable from the effects of a superposition of gravitational fields

$\hat{x}_A(t) = e^{\frac{i}{\hbar}\hat{H}_A t} \hat{x}_A e^{-\frac{i}{\hbar}\hat{H}_A t}$:

$$\hat{x}_A(t) = \hat{x}_A + \frac{\hat{P}_A}{m_A}t - \frac{1}{2}\frac{1}{m_A}\frac{dV(\hat{x}_A)}{d\hat{x}_A}t^2. \tag{21}$$

From this expression we come to the generalised $\hat{X}_A(t) = \hat{x}_A(t) - \hat{x}_A$, which describes the change in time of the position of A as compared to the initial position and replaces the function $X(t)$ in the extended Galilean transformation $\hat{U}_a$.

Following Eq. (7), the overall transformation $\hat{S}_{EP}$ reads

$$\hat{S}_{EP} = e^{-\frac{i}{\hbar}\left(\frac{\hat{\pi}_C^2}{2m_C} + \frac{m_C}{m_A}V(-\hat{q}_C)\right)t} \hat{\mathcal{P}}_{AC}^{(v)} \hat{Q}_t e^{\frac{i}{\hbar}\left(\frac{\hat{P}_A^2}{2m_A} + V(\hat{x}_A)\right)t}, \tag{22}$$

where the operator $\hat{\mathcal{P}}_{AC}^{(v)}$ was defined below Eq. (16) and the operator $\hat{Q}_t$ is defined as

$$\hat{Q}_t = e^{-\frac{i}{\hbar}\frac{m_B}{m_A}\left(\hat{P}_A - \frac{dV(\hat{x}_A)}{d\hat{x}_A}t\right)\hat{x}_B} e^{\frac{i}{\hbar}\left(\hat{P}_A - \frac{1}{2}\frac{dV(\hat{x}_A)}{d\hat{x}_A}t\right)\frac{\hat{p}_B}{m_A}t} e^{-\frac{i}{\hbar}\frac{m_B}{m_A^2}\int_0^t ds\left(\hat{P}_A - \frac{dV(\hat{x}_A)}{d\hat{x}_A}s\right)^2} \tag{23}$$

and represents the straightforward extension of the operator $\hat{U}_a$. Note that $\left\langle\frac{d^2\hat{x}_A}{dt^2}\right\rangle = -\left\langle\frac{d^2\hat{x}_A}{dt^2}\right\rangle$. Using the transformation in Eq. (22), the new Hamiltonian from the point of view of A is

$$\hat{H}_{BC}^{(A)} = \frac{\hat{\pi}_B^2}{2m_B} + \frac{\hat{\pi}_C^2}{2m_C} + \frac{m_C}{m_A}V(-\hat{q}_C) - \frac{m_B}{m_A}\frac{dV}{d\hat{x}_A}\Big|_{-\hat{q}_C}\hat{q}_B. \tag{24}$$

From Eq. (24), we can see that B evolves in a potential which is determined by the first derivative of the potential at the position $-\hat{q}_C$, while C moves in a potential given by the sum of $\frac{m_C}{m_A}V(-\hat{q}_C)$ and the interaction term involving its derivative. Hence, the quantum system B moves, in the reference frame of A, as if it were in a linear gravitational potential with a gravitational acceleration being an operator $\hat{g} = 1/m_A \, dV(\hat{x}_A)/d\hat{x}_A|_{-\hat{q}_C}$ in the Hilbert space of C. This is a formulation of the weak equivalence principle in QRF.

As an example, we will now apply $\hat{S}_{EP}$ on an arbitrary state $|\phi(t)\rangle_B$ of B and on a state $|\psi(t)\rangle_A$ of A, for which we assume that the two localised wave amplitudes were initially prepared as non-overlapping coherent states (i.e. minimum uncertainty wave-packets) with well defined position $x_i(0)$ and momenta $p_i(0)$, $i = 1, 2$. We evolve the state of A for time $t$ such that the two amplitudes still have

**Fig. 7** Representation of the conditions under which particle A effectively moves in a superposition of accelerations. The state of particle A and the piecewise linear potential $V(\hat{x}_A)$ are represented for the initial time 0 and for the time $t$ in two different yellow planes. At the initial time, in the background, the state is chosen to be a superposition of two coherent (Gaussian) states localised around $x_1$, with width $2\sigma_1$, and $x_2$, with width $2\sigma_2$. The individual states are localised within the spatial intervals that correspond to constant but different potential gradients. Under time evolution, the point where each of the two localised states is centred moves by $\delta x_i$, $i = 1, 2$, and the wave-packet spreads. For each of them, it is possible to identify a maximal time such that the individual localised states in the superposition still remain in the region where the gradient of the potential is constant. Up to this time A evolves in a superposition of accelerations

well defined position $x_i(t)$ and momenta $p_i(t)$, where the momentum at time $t$ is calculated analogously to $\hat{x}_A(t)$, i.e. $\hat{p}_A = \hat{P}_A - \frac{dV(\hat{x}_A)}{d\hat{x}_A}t$. We denote the state of each amplitude as $|\alpha_i(t)\rangle$, with $i = 1, 2$. Hence we obtain

$$\hat{S}_{EP}\frac{1}{\sqrt{2}}(|\alpha_1(t)\rangle_A + |\alpha_2(t)\rangle_A)|\phi\rangle_B = \frac{1}{\sqrt{2}}(|\alpha_1'(t)\rangle_C Q_t^1|\phi(t)\rangle_B + |\alpha_2'(t)\rangle_C Q_t^2|\phi(t)\rangle_B), \tag{25}$$

where the transformed coherent state $|\alpha_i'(t)\rangle_C$ is centred in $\left(-\frac{m_A}{m_C}x_i(t), -\frac{m_C}{m_A}p_i(t)\right)$ and $Q_t^i = e^{-\frac{i}{\hbar}\frac{m_B}{m_A}(p_i - m_A a_i t)\hat{x}_B} e^{\frac{i}{\hbar}\left(\frac{p_i t}{m_A} - \frac{a_i t^2}{2}\right)\hat{p}_B} e^{-\frac{i}{\hbar}\frac{m_B}{m_A^2}\int_0^t ds(p_i - m_A a_i s)^2}$. From A's point of view, B evolves in a superposition of gravitational accelerations, which is controlled by the state of C.

We conclude that we have a generalised form of the weak equivalence principle which holds when the reference frame is a quantum particle in superposition of accelerations. This analysis can be extended to a general potential $V(\hat{x}_A)$ acting for infinitesimal times, as we show in Supplementary Note 5.

The extension of the weak equivalence principle to quantum reference frames provides a good opportunity to test this framework in an experiment. In general terms, a suitable relative technique to verify the predictions of this section would be measuring relative degrees of freedom, for instance as done in refs. [36–39]. However, a specific proposal on how to do so goes beyond the scope of this work.

It is possible to recover the usual notion of the weak equivalence principle if the potential is linear in the entire space, i.e. $V(\hat{x}_A) = m_A a\hat{x}_A$. The generalised form of the displacement of the reference frame reads $\hat{X}_B(t) = \frac{\hat{p}_B}{m_A}t - \frac{at^2}{2}$ and the operator $\hat{S}_{EP}$ is analogous to Eq. (22) with $\hat{Q}_t = e^{-\frac{i}{\hbar}m_B\dot{\hat{X}}_A(t)\hat{x}_B} e^{\frac{i}{\hbar}\hat{X}_A(t)\hat{p}_B} e^{-\frac{i}{\hbar}\frac{m_B}{2}\int_0^t ds\dot{\hat{X}}_A^2(s)}$, where $\hat{X}_A(t)$ has been previously defined and the dot indicates the time derivative. The initial Hamiltonian $\hat{H}_{AB}^{(C)}$ in Eq. (19) is transformed to

$$\hat{H}_{BC}^{(A)} = \frac{\hat{\pi}_B^2}{2m_B} + \frac{\hat{\pi}_C^2}{2m_C} - m_C a\hat{q}_C - m_B a\hat{q}_B. \tag{26}$$

This result shows that the weak equivalence principle holds also if the reference frame is treated as a quantum system (and can therefore be delocalised) with its own dynamics.

**Application: notion of rest frame of a quantum system.** In this section, we show how our formalism enables us to define the notion of rest frame when the system is in a superposition of momenta from the point of view of the initial laboratory frame. The rest frame of a system is the frame of reference in which the system is at rest. Physical laws standardly take a simple form in the rest frame; for example, the rest frame Hamiltonian gives the dynamics of the internal degrees of freedom (e.g. spin). It is therefore useful to know how to map the descriptions in the rest and the laboratory frames of reference. As long as the system moves along a

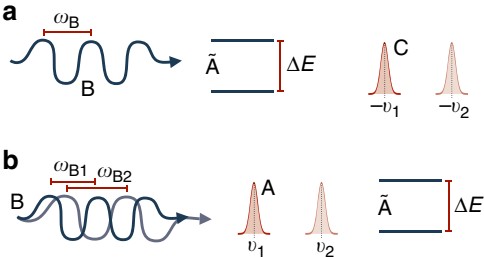

**Fig. 8** Transformation to the rest frame of a quantum system. The interaction between a photon B and the internal degrees of freedom Ã of an atom as described from the point of view of **a** the rest frame of the atom A itself and **b** the laboratory C. We consider the situation when the atom does not have a sharp momentum in the laboratory reference frame and calculate which state of the photon and the atom we have to prepare to maximise the probability of absorbing the photon. The description of the situation is simplest in the rest reference frame (**a**). If the photon has spectral frequency $\omega_B = \frac{\Delta E}{\hbar}$ corresponding to the atom's energy gap, the probability is maximised. For simplicity of illustration, the state of the laboratory is described as a superposition of two amplitudes sharp around the velocities $-v_1$ and $-v_2$. From the point of view of C, this situation is described as in **b**, in which the state of the photon and the external degrees of freedom of the atom are entangled. For each velocity $v_i$, $i = 1, 2$ of atom A, the frequency of the photon is Doppler-shifted as: $\omega_{Bi} = \omega_B\left(1 - \frac{v_i}{c}\right)$ for $i = 1, 2$. The measurement of the frequency $\hat{\omega}_B$ in A's reference frame, corresponds to the measurement of the observable $\hat{\omega}_B\left(1 + \frac{\hat{p}_A}{cm_A}\right)$ in the laboratory reference frame (see the next section). This ensures that if the photon is found absorbed ('detected') by the atom in its RF, so it will in the laboratory RF

classical trajectory and is not treated as a dynamical degree of freedom the map can be achieved through a coordinate transformation between the two reference frames. However, in quantum mechanics, a system can evolve in a superposition of classical trajectories. How can we 'move' to the rest frame of a particle that is in superposition of momenta with respect to the laboratory reference frame? Here, by working out an explicit example, we show how our formalism can be used to recover the notion of the rest frame of a quantum system, when the semiclassical approximation fails.

We consider the situation illustrated in Fig. 8, in which an atom with its external (A) and internal (Ã) degrees of freedom interacts with a photon (B), as seen from the laboratory reference frame (C). We assume the internal degrees of freedom to be internal energy states of a two-level system. We want to find conditions under which the photon is in resonance with the internal energy levels of the atom from different frames of reference. More precisely, we want to find the state of the atom and photon such that the probability for the transition is maximised, in the case when the atom does not have a well-defined momentum in the laboratory reference frame. We know that when the source of a photon (where the source is at rest from the point of view of the laboratory reference frame) and the receiver (i.e. the atom) are in relative motion towards each other, and in the limit of small relative velocity between emitter and receiver, the frequency is Doppler-shifted according to $\omega'_B = \omega_B\left(1 + \frac{v}{c}\right)$, where $\omega_B$, $\omega'_B$ are respectively the emitted and received frequency of the photon, $v$ is the relative velocity between emitter and receiver, and $c$ is the speed of light in the medium.

The condition for the absorption of the photon is simplest in the rest frame of A. Suppose that in this frame the entire state is given as $|\psi_t\rangle_C|1, \omega_B\rangle_B|g\rangle_{\tilde{A}}$. Here, the state of the laboratory at time $t$ is $|\psi_t\rangle_C = \int d\pi_C e^{-\frac{i}{\hbar}\frac{\pi_C^2}{2m_C}t}\psi\left(-\frac{m_A}{m_C}\pi_C\right)|\pi_C\rangle_C$ (this is related to the momentum distribution of atom A in laboratory reference frame C), $|g\rangle_{\tilde{A}}$ is the ground state of the internal degrees of freedom of the atom and $|1, \omega_B\rangle_B$ is the 1-photon state of B with frequency $\omega_B = \frac{\Delta E}{\hbar}$, where $\Delta E$ is the energy gap between the ground and the excited state of the internal energy. The Hamiltonian in the rest frame is taken to be $\hat{H}^{(A)}_{\tilde{A}BC} = \frac{\hat{\pi}_C^2}{2m_C} + \hbar\hat{\omega}_B + \hat{H}_{\tilde{A}}$, where $\hbar\hat{\omega}_B$ is a simplified photon Hamiltonian in the one-particle sector. Here, we promote the frequency $\omega_B$ to an operator because the frequency shift due to the Doppler effect changes the mode of the photon state, but leaves the number of particles invariant. A more complete description of the Hamiltonian would involve the creation and annihilation operators, but we omit it here because it does not influence our results. The frequency operator $\hat{\omega}_B$ acts on the single-photon Hilbert space and is such that $\hbar\hat{\omega}_B|\omega_B\rangle_B = \hbar\omega_B|\omega_B\rangle_B$, where the usual relation between momentum and

frequency holds, i.e. $\hbar\omega_B = c|\pi_B|$. Finally, $\hat{H}_{\tilde{A}} = E_g|g\rangle_{\tilde{A}}\langle g| + E_e|e\rangle_{\tilde{A}}\langle e|$ is the Hamiltonian of the internal degrees of freedom, with $E_e - E_g = \Delta E$.

To change the reference frame, we apply a boost transformation between A and C, and the transformation which gives the Doppler shift on the photon. Overall, we obtain

$$\hat{S}_D = e^{-\frac{i}{\hbar}\frac{\hat{p}_A^2}{2m_A}t}\hat{\mathcal{P}}^{(v)}_{CA}\hat{R}^B_{f_C(\hat{\pi}_C)}e^{\frac{i}{\hbar}\frac{\hat{\pi}_C^2}{2m_C}t}, \tag{27}$$

where the operator $\hat{\mathcal{P}}^{(v)}_{CA}$ is the adjoint of $\hat{\mathcal{P}}^{(v)}_{AC}$ defined after Eq. (16), i.e. $\hat{\mathcal{P}}^{(v)}_{CA} = \left(\hat{\mathcal{P}}^{(v)}_{AC}\right)^\dagger$, and $\hat{R}^B_{f_C(\hat{\pi}_C)} = \exp\left(\frac{i}{\hbar}\log\sqrt{f_C(\hat{\pi}_C)}(\hat{q}_B\hat{\pi}_B + \hat{\pi}_B\hat{q}_B)\right)$, with $f_C(\hat{\pi}_C) = 1 + \frac{\hat{\pi}_C}{cm_C}$. Specifically, the operator $\hat{R}^B_{f_C(\hat{\pi}_C)}$ represents the Doppler shift of the photon. Finally, the transformation between the spatial degrees of freedom of A and C is the boost transformation in Eq. (16). We obtain

$$\hat{S}_D\hat{\pi}_C\hat{S}_D^\dagger = -\frac{m_A}{m_C}\hat{p}_A; \qquad \hat{S}_D\hat{\pi}_B\hat{S}_D^\dagger = \left(1 + \frac{\hat{p}_A}{m_A c}\right)\hat{p}_B. \tag{28}$$

Applying this transformation to the Hamiltonian $\hat{H}^{(A)}_{\tilde{A}BC}$ yields

$$\hat{H}^{(C)}_{A\tilde{A}B} = \frac{\hat{p}_A^2}{2m_A} + \hat{H}_{\tilde{A}'} + \hbar\hat{\omega}_B\left(1 + \frac{\hat{p}_A}{m_A c}\right), \tag{29}$$

where $\hat{\omega}_B = \frac{c}{\hbar}|\hat{p}_B|$. From the perspective of the laboratory C, the Hamiltonian entangles the momentum of the atom A with the frequency of the photon, while the internal degrees of freedom are unchanged. The state of the joint system of the atom, with its internal and external degrees of freedom, and the photon is

$$|\Psi\rangle^{(C)}_{A\tilde{A}B} \propto \int dp_A e^{-\frac{i}{\hbar}\frac{p_A^2}{2m_A}t}\psi(p_A)|p_A\rangle_A|1, \omega_B\left(1 - \frac{p_A}{m_A c}\right)\rangle_B|g\rangle_{\tilde{A}}. \tag{30}$$

The state (30) is the one which has to be prepared in the laboratory reference frame to maximise the absorption probability. We see that the frequency of the photon B is Doppler shifted by an amount that depends on the velocity of the atom A. In the next section we show that, by mapping the observables in reference frame A to those in reference frame C, the absorption of the photon is predicted consistently in both reference frames, i.e. if the photon is detected in A's reference frame, so it will in C's reference frame.

**Measurements as seen from a quantum reference frame**. In this section we analyse how a measurement procedure performed in one QRF looks like as seen from another QRF. We assume that an observer in reference frame C performs a measurement on the quantum systems A and B. How does an observer in the reference frame A describe this procedure? Note that the procedure in general includes also a measurement on the reference frame of A itself. This situation differs from the Wigner-friend scenario, in which one observer (friend) performs a measurement, while the other (Wigner) considers the process to be unitary[24,41]. In the present case both observers agree that a measurement is performed, though, as we will see, they might have a different view on which systems and which measurement is performed.

Consider that in C's reference frame observable $\hat{O}^{(C)}_{AB}$ is measured. The transformed observable in A's reference frame is $\hat{O}^{(A)}_{BC} = \hat{S}\hat{O}^{(C)}_{AB}\hat{S}^\dagger$, where $\hat{S}$ is a general operator which implements the transformation from the reference frame of C to the reference frame of A. Using the cyclicity of the trace, it is immediate to verify that

$$\langle\hat{O}^{(C)}_{AB}\rangle = \text{Tr}_{AB}(\hat{\rho}^{(C)}_{AB}\hat{O}^{(C)}_{AB}) = \text{Tr}_{BC}(\hat{\rho}^{(A)}_{BC}\hat{O}^{(A)}_{BC}) = \langle\hat{O}^{(A)}_{BC}\rangle, \tag{31}$$

where $\hat{\rho}^{(A)}_{BC}$ is the quantum state of B and C relative to A. An explicit example, using operator $\hat{S}_x$ in Eq. (2), is a measurement of the position operator $\hat{q}_B$ of the quantum system B in the reference frame of A, which is equivalent to the measurement of $\hat{x}_B - \hat{x}_A$ in the reference frame of C.

To make these statements more concrete, we adopt a measurement scheme (see, for instance[42]) and check how the measurement procedure transforms when we change reference frame. The measurement procedure in C's and A's reference frames is depicted in Fig. 9. The measurement scheme consists in adding an ancillary system consisting of a pointer in the state $\xi_E \in \mathcal{H}^{(C)}_E$ and of external (position) degrees of freedom of the measurement apparatus in the state $\sigma_M \in \mathcal{H}^{(C)}_M$. The measurement of the observable $\hat{O}^{(C)}_{AB}$ on the quantum system $\hat{\rho}^{(C)}_{AB}$ can be then described as an interaction between the pointer and the quantum system, followed by a projection in the Hilbert space of the pointer. The probability of measuring the outcome $b^*$ is

$$p(b^*) = \text{Tr}_{AB}\left[\hat{\rho}^{(C)}_{AB}\hat{O}^{(C)}_{AB}(b^*)\right] = \text{Tr}_{ABEM}\left[\mathcal{C}(\hat{\rho}^{(C)}_{AB} \otimes \xi_E) \otimes \sigma_M(1_{ABM} \otimes \hat{F}_E(b^*))\right], \tag{32}$$

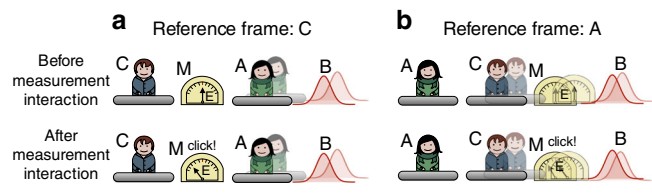

**Fig. 9** Measurement in quantum reference frames. The measurement procedure as seen from the point of view of C, in **a**, and of A, in **b**. In **a**, C prepares an ancillary system, constituted by the position degrees of freedom of the apparatus M and a pointer E. The ancilla interacts with the quantum systems A and B. Subsequently, a projective measurement of the state of the pointer gives the outcome. In **b**, A describes the initial state as an entangled state of B, C, and M. The pointer E then interacts with both systems B and C. Finally, system E is measured. The measurement probability is the same as in the reference frame of C

where $\mathcal{C}$ is a unitary channel entangling the states from the Hilbert spaces of A and B with those of the pointer E, and $\hat{F}_E$ is a projector on $\mathcal{H}_E^{(C)}$.

We measure the outcome $b^*$ as $\hat{O}_{AB}^{(C)}(b^*) = |b^*\rangle_{AB}\langle b^*|$. Then, if we choose $\hat{F}_E(b^*) = |b^*\rangle_E\langle b^*|$, $\xi_E$ such that $|\xi_E(x_E)|^2 = \delta(x_E)$ and

$$\mathcal{C}(\hat{\rho}_{AB}^{(C)} \otimes |\xi\rangle_E\langle\xi|) = C_{ABE}(\hat{\rho}_{AB}^{(C)} \otimes |\xi\rangle_E\langle\xi|)C_{ABE}^\dagger, \tag{33}$$

where $C_{ABE} = \exp\left(-\frac{i}{\hbar}\hat{O}_{AB}^{(C)}\hat{p}_E\right)$, the condition (32) is satisfied for all $\hat{\rho}_{AB}^{(C)}$. Notice that by $\hat{x}_E$ and $\hat{p}_E$ we mean the position of the pointer in some abstract space of the internal degrees of freedom of the apparatus, not the real position (relative to C) of the measurement apparatus in space.

We now turn to the description from the point of view of A. For concreteness, we consider the transformation between two frames of reference C and A to be the map in Eq. (2); the formalism can be straightforwardly generalised to other maps. Considering the degrees of freedom of the ancilla the map is modified to include the external degrees of freedom of the measurement apparatus, i.e., $\hat{S}_{x,M} = \hat{\mathcal{P}}_{AC}e^{\frac{i}{\hbar}\hat{x}_A(\hat{p}_B+\hat{p}_M)}$, while the degrees of freedom of the pointer are considered as translational invariant, and therefore not transformed. From A's reference frame the measurement process is

$$p(b^*) = \text{Tr}_{BCEM}\left[C_{BCEM}^{(A)}(\hat{\rho}_{BCM}^{(A)} \otimes \xi_E)(C_{BCEM}^{(A)})^\dagger(1_{BCM} \otimes \hat{F}_E(b^*))\right], \tag{34}$$

where the state $\hat{\rho}_{BCM}^{(A)} = \hat{S}_{x,M}(\hat{\rho}_{AB}^{(C)} \otimes \sigma_M)\hat{S}_{x,M}^\dagger$ becomes entangled with the external degrees of freedom of the measurement apparatus, and $C_{BCEM}^{(A)} = \hat{S}_{x,M}C_{ABE}\hat{S}_{x,M}^\dagger = \exp\left(-\frac{i}{\hbar}\hat{O}_{BC}^{(A)}\hat{p}_E\right)$. We see that the two observers in the reference frames C and A disagree on which systems undergo the measurement and which observables are measured. For observer C, systems A and B are measured with the help of an ancilla E whose internal and external degrees of freedom are initially in a state that factorizes out. For observer A, systems B and C are measured via the ancilla whose external (but not internal) degrees of freedom are initially entangled with C. Therefore, a measurement model for C is transformed into a measurement model for A.

Notice that the measurement procedure just considered is different from A performing a measurement in her reference frame. In the previous paragraphs, when we changed the reference frame from C to A, we still described the measurement performed by C from the point of view of A. Clearly, A can apply the same measurement procedure as C, with the observables defined in her reference frame $\hat{O}^{(A)}$. A concrete example of this situation we consider the atom-photon interaction from the previous section. We make the following identification: the atom's external degree of freedom is system A, the photon is system B, the atom's internal degrees of freedom Ā are ancilla E, and finally the laboratory is system C. We consider that A 'measures' the frequency of the photon by transition of its internal level from the ground to the excited state. This transition can happen only if the frequency of the photon matches the energy gap of the atom. How is this condition written in the laboratory reference frame, in which the photon frequency is Doppler-shifted?

In the rest frame of the atom A, the channel $C_{B\bar{A}}^{(A)}(\cdot) = C_{B\bar{A}}^{(A)}(\cdot)C_{B\bar{A}}^{(A)\dagger}$ entangling the states of the Hilbert spaces of B with those of Ā is such that $C_{B\bar{A}}^{(A)} = |e\rangle_{\bar{A}}\langle g| \otimes |0, \omega\rangle_B\langle 1, \omega| + h.c.$, and the project or $\hat{F}_{\bar{A}}^{(A)} = |e\rangle_{\bar{A}}\langle e|$. Changing to the laboratory frame C via the application of the $\hat{S}_D$ operator in Eq. (27), the entangling channel becomes $C_{AB\bar{A}}^{(C)} = 1_A \otimes |e\rangle_{\bar{A}}\langle g| \otimes \hat{R}_{f_A(-\hat{p}_A)}^B|0, \omega\rangle_B\langle 1, \omega|\hat{R}_{f_A(-\hat{p}_A)}^{B\dagger} + h.c.$, with $f_A(-\hat{p}_A) = 1 - \frac{\hat{p}_A}{cm_A}$, while the projector is unchanged, i.e. $\hat{F}_{\bar{A}}^{(A)} = \hat{F}_{\bar{A}}^{(C)}$. This ensures,

by construction, that the probabilities of the photon being absorbed are the same in the reference frame A and C. What changes is the measured observable in the two reference frames: in the rest frame A one measures the frequency $\hat{\omega}_B$, while in the laboratory frame the measurement involves both the external degrees of freedom of the atom and the photon. In this case, the observable is $\left(1 + \frac{\hat{p}_A}{cm_A}\right)\hat{\omega}_B$.

**Comparison of previous approaches to quantum reference frames**. In this Section, we compare different approaches to the topic of quantum reference frames, emphasising differences and similarities to our approach.

Much work has been done on the subject of QRFs starting from the seminal papers by Aharonov and Susskind[1,2] and Aharonov and Kaufherr[3]. In refs. [1,2] the authors established a relation between superselection rules and the lack of a frame of reference. The authors then challenged the existence of superselection rules via some examples, where the superselection rule could be overcome by introducing a reference frame that was correlated with the system. The simplest example of this is described in ref. [4]. There it is shown that, if two observers do not share information about their relative phase, this implies a superselection rule for photon number. This superselection rule can be overcome by introducing an appropriate quantum reference frame which is entangled with the system in such a way that the total photon number is conserved. In ref. [3] it was shown that it is possible to consistently formulate quantum theory without appealing to classical reference frames as well-localised laboratories of infinite mass.

QRFs have been considered as resources in quantum information protocols and quantum communication in refs. [4–12]. These works mainly focus on (a) the consequences of the lack of a shared reference frame for quantum information tasks, on (b) the generalisation of the fact that superselection rules can be overcome by choosing an appropriate quantum system as a reference frame, and on (c) "bounded" reference frames. This means that, in a quantum communication protocol where a system is sent from A to B, the final reference frame only possesses limited or no information on the initial reference frame. In order to obviate this problem, most approaches resort to an encoding of quantum information in the relational degrees of freedom (see, e.g. ref. [5]). The tool used to achieve this encoding is the $\mathcal{G}$-twirl operation, which consists in an average over the group of symmetries of the external reference frame $G$. This operation consists in expressing the quantum state of the system under study, $\rho$, in a way that does not contain any information about the external reference frame. The $\mathcal{G}$-twirl operation is mathematically expressed as $\mathcal{G}(\rho) = \int d\mu(g)U(g)\rho U^\dagger(g)$, where $U(g)$ is the unitary representation of the group element $g \in G$, and $\mu(g)$ is the group-invariant measure. This approach shares some methodological similarities with our work, such as the relevance of the relational degrees of freedom; however, there are some important differences. Firstly, we do not assume the existence of an external reference frame, whose degrees of freedom need to be averaged out. This means that, differently to the other approaches mentioned, we do not apply techniques, such as the $\mathcal{G}$-twirling, in order to find the relational quantities, because our formalism is genuinely relational from the very start. Secondly, we do not address the problem of communication in absence of a shared reference frame. Finally, we assume that the relation between the initial and final reference frame is known and hence always given by a unitary transformation. Moreover, reference frames in our approach are not bounded; they allow to assign quantum states to all external systems to arbitrary precision.

Other authors focus on the possible role of QRFs in quantum gravity[13], or point out how QRFs, together with a relational approach, can lead to intrinsic decoherence due to the finite size of the systems considered[14,15]. Considering reference frames quantum mechanically is a fundamental ingredient in formulating relational quantum theory, which makes no use of an external reference frame to specify its elements[23]. A relational approach to QRFs has been also considered in refs. [16–18], where the limit to an absolute reference frame was formalised, and in ref. [19], where a symmetry group for transformations of a spin system was reconstructed.

Among the works listed, special mention should be paid to ref. [8], which is the closest to our work. There, the $\mathcal{G}$-twirl operation is introduced to average over all the external information to the joint system of two particles, A and S, one of which (A) serves as a reference frame. After this operation, the only quantities remaining are the relative variables of the system S with the quantum reference frame. This description that the QRF A gives of the system presents some similarities with ours. The main differences, at this level, are that in our method, after the transformation, we describe the initial reference frame in addition to the second particle, and we do not rely on any external frame. The rest of the paper, addressing the change to a different QRF, asks a different question to us, and thus arrives to a different result. In ref. [8], the authors switch to the description of a third quantum system B, whose relationship with the initial QRF is not known and will be acquired through a quantum measurement. To this end, they propose a protocol, consisting of multiple steps: (1) the quantum state of the QRF B, initially factored out from the system and the QRF A, undergoes a $\mathcal{G}$-twirl operation, which removes the information about the external frame; (2) a quantum measurement on the joint state of the QRFs A and B is performed to establish the relation between the two QRFs; 3) The initial QRF A is discarded. The result of this series of operation is the relational description of the state of the system S with respect to B. As a result of the measurement, the group-averaging operation, and the discarding of the initial QRF the transformation results in a decohered state of the system in general. The

appearance of decoherence is a fundamental difference with our approach, where we assume that the relative description of the two QRFs (in our language, the state of the new QRF from the point of view of the old one) is known and hence the change of reference frame is unitary. In addition, in ref. [8] the system and the new QRF are never entangled at the beginning of the protocol, while our approach does not have this restriction.

Along the lines of[1–3], the subject of quantum reference frames has been recently revised by Angelo and coworkers[20–22], and fundamental contributions were provided to understand reference frames as quantum-mechanical systems. In these works, one methodologically begins by defining the state in an external frame, and then moves to the centre of mass and relative coordinates, tracing out the centre of mass coordinate as a degree of freedom that describes its position in this external frame. It was claimed that the equations of motion from the perspective of such relative degrees of freedom are compatible with Galilean relativity and the weak equivalence principle, but that the Hamilton formalism is not, as no Hamiltonian can be found that only depends on the coordinates accessible to the quantum frame of reference [22]. This constitutes a fundamental difference with our work, where the Hamiltonian formalism can always be used. The main difference resides in the fact that our transformation is canonical, characteristics which automatically guarantees that a Hamiltonian system is transformed to another Hamiltonian system.

Differently to other approaches, our formalism is genuinely relational by construction. This means that, while the focus of the previous works cited has been to obtain the relational degrees of freedom, we consider from the very start physical degrees of freedom to be relational from the point of view of a chosen QRF. Moreover, similarly to ref. [8] and to refs. [20–22], we abandon the view that reference frames are abstract entities, which are useful to fix a set of coordinates, and instead treat the reference frame in the same way as any physical system, featuring its physical state and dynamics. Therefore, a QRF has a quantum state and a Hamiltonian relative to another QRF, and the latter QRF has a quantum state and a Hamiltonian relative to the former QRF. Our paper formalises the transformation of states, dynamics and measurements between these two QRFs, using exclusively relational quantities.

Every quantum state, as specified relative to a QRF, encodes the relational information in terms of probabilities for measuring all the degrees of freedom external to the QRF. As a consequence, our formalism does not appeal to an absolute reference frame and consequently does not require the existence of an 'external' perspective. Moreover, differently to other approaches in the literature, our work is not about the lack of a shared reference frame, and we do not consider our QRFs to be "bounded", feature which usually leads to an imprecise state assignment or to a noisy measurement result. In contrast, our QRFs allow to make state assignments to external systems with arbitrary precision. We adopt an operational approach assuming that every QRF is equipped with hypothetical devices that allow for an operational justification of such state assignments. This operational view, indeed very useful, does not require to have laboratories (and possibly observers) in macroscopic superpositions. We will exemplify the relevance of our formalism for quantum particles by applying it to 'move' to the rest frame of a particle that is in a superposition of momenta with respect to the laboratory frame and has internal degrees of freedom that can serve as a 'measurement device'. Possible tests of our framework would involve experimental techniques such as, for instance, those in refs. [36–39], which are able to probe the relative degrees of freedom.

## Data availability
Data sharing not applicable to this article as no datasets were generated or analysed during the current study.

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

## Acknowledgements

We thank Alessio Belenchia, Fabio Costa, and Philipp Höhn for useful discussions. We acknowledge the support of the Austrian Science Fund (FWF) through the Doctoral Programme CoQuS (Project no. W1210-N25) and the projects I-2526-N27 and I-2906, as well as the research platform TURIS. This project has received funding from the European Union's Horizon 2020 research and innovation programme under the TEQ collaborative project (Grant agreement no. 766900). This work was funded by a grant from the Foundational Questions Institute (FQXi) Fund. This publication was made possible through the support of a grant from the John Templeton Foundation. The opinions expressed in this publication are those of the authors and do not necessarily reflect the views of the John Templeton Foundation.

## Author contributions

F.G., E.C.R. and Č.B. contributed to all aspects of the research with the leading input from F.G.

## Additional information

**Competing interests:** The authors declare no competing interests.

