## [Peer Review File · Nature Communications]

Reviewers' comments:

Reviewer #1 (Remarks to the Author):

This manuscript studies the problem of describing quantum systems relative to reference frames which are themselves quantum mechanical.

The state of any physical system is defined only relative to some reference frames. Often we treat these reference frame as external abstract systems, and ignore their possible noise and quantum fluctuations. But, obviously, this is just an idealization and actual clocks and reference frames are themselves quantum mechanical. In other words, the state of a quantum system should be defined based on its relation with another quantum system. The current manuscript studies this problem and proposes a framework for describing the state of quantum systems relative to quantum reference frames. Furthermore, it explains how the covariance of dynamical physical laws can be stated relative to quantum reference frames.

I found the paper interesting and insightful. However, I have several concerns about both the result and the presentation, and would like to postpone my final decision, until I hear the authors responses. Please see the details below:

1. In the last 20 years there have been a lot of progress in understanding quantum reference frames and relational degrees of freedom. For instance, different groups of researchers have studied the problem of describing quantum systems relative to bounded size quantum reference frames. Some important works, which are closely related, are not cited in this paper. E.g. Phys.Rev. A69 (2004) 052326 by Kitaev, Mayers and Preskill, New J. Phys. 11, 063013 (2009) by Bartlett, Rudolph, Spekkens, and Turner.

2. In my opinion, the authors need to compare their approach with the previous approaches to this problem. For instance, Ref 4 which is a review paper on this topic (published almost ten years ago in Rev of Modern Phys) has sections on "Quantum treatment of reference frames" and "Relational descriptions of phase". Also, in a more recent paper titled "Changing quantum reference frames" (Ref 7), this problem is studied from a slightly different point of view.

How the approach taken in the current manuscript relates to these previous works?

3. The authors say “..notions of superposition and entanglement are reference-frame dependent.Here we show that the effect can arise due to genuine quantum relationships between reference frames even in non-relativistic quantum mechanics. ”

The fact that in non-relativistic scenarios entanglement and superposition can be reference frame-dependent, has been previously observed and studied. See, for instance, Phys. Rev. Lett. 91, 097903 (2003) by Bartlett and Wiseman, and other references in the review paper mentioned above (Ref 4).

4. In formula 2 the authors introduce a recipe for transforming the bi-parite state associated to the system B and the old reference frame C to another bipartite state associated to the system B and the new reference frame A. They claim this is a “natural” choice.

But I do not understand what is the physical meaning of this transformation. Is this formula in some sense “unique”? If not, then why we should use this formula, and what are the alternatives?

I have several other questions about this formula.

i) The transformation in Eq.(2) is defined only when A and C have the same Hilbert spaces. But, what if this is not the case? For instance, what happens if the new reference frame is a composite system?

ii) Suppose the old reference frame C, is very “noisy” , e.g. it is in the totally mixed state, or a high temperature state, and the new reference frame A is an ideal reference frame. The observer who uses the noisy reference frame cannot have a precise description of system B. So, why the observer who uses the ideal reference frame A should care about this description. In other words, why the new observer should care about the state relative to the old reference frame C and use formula in Eq.(2)?

iii) If we start with reference frame C, then go to reference frame A and then go back to reference frame C again, in general, we do not get back the same state. Why this is the case? Which description is the valid description of system B relative to reference frame C?

iv) Suppose instead of just two reference frames A and C there are many other quantum reference frames. Then, how the formula is modified? Does one need to consider all the other quantum reference frames in the world to describe system B relative to a quantum reference frame A?

v) At various places the authors claim that their approach is operational and does not use any external "absolute" reference frame. But, when in formula 2, they right observables x_A and p_B , these observables cannot be measured (defined) unless one has an ideal reference frame. In other words, using a (noisy) bounded quantum reference frame, one cannot measure observables x_B and p_b exactly. It seems that to define/measure these observables one need an absolute ideal reference frame.

Reviewer #2 (Remarks to the Author):

Please see the attached pdf.

Please note that the changes from the initial version are marked out in blue in the revised version of the manuscript.

Point-by-point reply to Referee #1

We thank the Referee for their comments. They are interesting and pertinent, and we included them in the manuscript. Below we write a point-by-point reply to the Referee's comments.

This manuscript studies the problem of describing quantum systems relative to reference frames which are themselves quantum mechanical.

The state of any physical system is defined only relative to some reference frames. Often we treat these reference frame as external abstract systems, and ignore their possible noise and quantum fluctuations. But, obviously, this is just an idealization and actual clocks and reference frames are themselves quantum mechanical. In other words, the state of a quantum system should be defined based on its relation with another quantum system. The current manuscript studies this problem and proposes a framework for describing the state of quantum systems relative to quantum reference frames.

Furthermore, it explains how the covariance of dynamical physical laws can be stated relative to quantum reference frames.

I found the paper interesting and insightful. However, I have several concerns about both the result and the presentation, and would like to postpone my final decision, until I hear the authors responses. Please see the details below:

1. In the last 20 years there have been a lot of progress in understanding quantum reference frames and relational degrees of freedom. For instance, different groups of researchers have studied the problem of describing quantum systems relative to bounded size quantum reference frames. Some important works, which are closely related, are not cited in this paper. E.g. Phys.Rev. A69 (2004) 052326 by Kitaev, Mayers and Preskill, .

While we find these references to be relevant for putting our work in a broader context, we note that they consider a different research question. In particular, our work is not about bounded-size quantum reference frames, but about (a) reference frames from which one can make quantum state assignments with arbitrary precision, and (b) which could be in superposition with respect to each other. For instance, a reference frame A described, from the point of view of another observer C, by a single ket $|x\rangle$ is not bounded as it enables assigning a well-defined position to another system B. Furthermore, a state $|x\rangle + |x'\rangle$ of A given by a spatial superposition of two kets as seen from C enables again assigning, for each amplitude, a precise relative position to the external system B. In short, it is not the consequences of the reference frames having bounded size that we explore in this work, but rather the consequences of the reference frame being in superposition or entangled. We would like to emphasise this point right from the beginning, because it will help to clarify some of the Referee's further comments.

Nonetheless, we thank the Referee for pointing out these two works to us, because they are relevant to the literature of bounded-size reference frames and communication protocols, which we refer to. We added the citation of the two works mentioned when referring to bounded reference frames and revised the Introduction correspondingly.

2. In my opinion, the authors need to compare their approach with the previous approaches to this problem. For instance, Ref 4 which is a review paper on this topic (published almost ten years ago in Rev of Modern Phys) has sections on "Quantum treatment of reference frames" and "Relational

descriptions of phase". Also, in a more recent paper titled "Changing quantum reference frames" (Ref 7), this problem is studied from a slightly different point of view.

How the approach taken in the current manuscript relates to these previous works?

The review paper published in Rev. Mod. Phys. largely focusses on the lack of a reference frame and on bounded-size reference frames. The paper "Changing quantum reference frames" provides an operational approach to correlate the description of two quantum systems used as a reference. All the mentioned approaches share some methodological similarities with ours, but ask different questions and have different goals, as mentioned in our answer to point 1). In order to put our work into context, we added the Appendix "Comparison between different approaches to quantum reference frames" (Appendix A). Here, we discuss in detail the differences and similarities between our approach and the existing ones.

3. The authors say "...notions of superposition and entanglement are reference-frame dependent.Here we show that the effect can arise due to genuine quantum relationships between reference frames even in non-relativistic quantum mechanics."

The fact that in non-relativistic scenarios entanglement and superposition can be reference frame-dependent, has been previously observed and studied. See, for instance, Phys. Rev. Lett. 91, 097903 (2003) by Bartlett and Wiseman, and other references in the review paper mentioned above (Ref 4).

We thank the Referee for pointing this out to us. We haven't been clear on this point. In previous literature (including the works mentioned by the Referee), the dependence of entanglement and superposition on the reference frame is due to the appearance of decoherence, which enters as a consequence of the G-twirl operation. This operation is performed when there is a lack of reference frame. In our work, in contrary to other approaches, the transformation from one quantum reference frame to another is unitary, so no decoherence arises. Yet, as we show, entanglement and superposition are still reference frame dependent. In order to show that this frame dependence is of a different nature to the one considered in previous works, we have rephrased the corresponding paragraph on p. 10 to improve the clarity of our statement and added the Appendix "Comparison between different approaches to quantum reference frames" (Appendix A), where we analyse similarities and differences between our approach and others in the literature.

4. In formula 2 the authors introduce a recipe for transforming the bi-parite state associated to the system B and the old reference frame C to another bipartite state associated to the system B and the new reference frame A. They claim this is a "natural" choice.

But I do not understand what is the physical meaning of this transformation. Is this formula in some sense "unique"? If not, then why we should use this formula, and what are the alternatives?

The transformation in Eq. (2) can be seen as the quantum-mechanical extension of the transformation between two classical reference frames. This extension *is* unique in the Lagrangian formalism (as mentioned on p. 8 of the revised version), and unique in the Hamiltonian formalism when the transformation to relative coordinates is required to be "canonical, linear in phase-space observables, and does not mix coordinates and momenta," (p. 8 of the manuscript).

One of the main results of our work is that it has no meaning to define **the** quantum

reference frame associated to a physical system, but we rather have to specify the relational variables we are interested in. The transformation in Eq. (2) is unique if we are interested in relative coordinates. If, alternatively, we are interested in a transformation to relative momenta, such as on p. 8, or in the choice of a different physical transformation, such as the boost transformation or the transformation to an accelerated quantum reference frame in Sections III B and IV, the transformation is different to the one in Eq. (2). Once the transformation is fixed, i.e. once the relational variables and the type of transformation (boost, translation, etc) are chosen, the S operator is uniquely defined. We have changed the corresponding part of the paper to clarify this question (last paragraph on p. 6 to the beginning of p.8).

I have several other questions about this formula.

i) The transformation in Eq.(2) is defined only when A and C have the same Hilbert spaces. But, what if this is not the case? For instance, what happens if the new reference frame is a composite system?

This is an interesting and important question. In our view, the question whether the system is composite or not is not essential, as long as the dimensions of the Hilbert spaces are the same. The reason for this is that we want to ensure that the transformation is unitary. In the paper, we consider infinite-dimensional Hilbert spaces, which are all isomorphic. Therefore, there is always the same number of relational degrees of freedom, no matter in which reference frame we describe the external systems. However, it can happen that the Hilbert spaces are of different and finite dimensions. In this case, it might appear that there is loss of information (coherence) when moving from the Hilbert space of larger to the one of smaller dimension. However, this point is beyond the scope of the present manuscript. We added a comment on p. 8 of the revised version of the manuscript.

ii) Suppose the old reference frame C , is very "noisy" , e.g. it is in the totally mixed state, or a high temperature state, and the new reference frame A is an ideal reference frame. The observer who uses the noisy reference frame cannot have a precise description of system B . So, why the observer who uses the ideal reference frame A should care about this description. In other words, why the new observer should care about the state relative to the old reference frame C and use formula in Eq.(2)?

We thank the Referee for this comment. Note that, in our framework, the reference frame can be in a noisy state only with respect to another reference frame. For instance, C can be noisy with respect to A . When system C is noisy from the point of view of a given reference frame A , after a transformation to the new reference frame (say again C), it is the initial frame A to be noisy. This example shows again that our approach differs from the view of quantum reference frames as potentially noisy resources for performing quantum operations. In our approach, we always assume unbounded resources in this respect.

iii) If we start with reference frame C , then go to reference frame A and then go back to reference frame C again, in general, we do not get back the same state. Why this is the case? Which description is the valid description of system B relative to reference frame C ?

The claim made by the Referee is incorrect in this point. The transformation in Eq.

(2) satisfies the transitive property, which implies that the transformation obtained composing the change from C to A and then from A to C gives the identity operator. We added a new Appendix, "Transitivity of the reference frame transformation" (Appendix C), where we prove this result.

iv) Suppose instead of just two reference frames A and C there are many other quantum reference frames. Then, how the formula is modified?

If there are more quantum reference frames, the same method applies, with the same transformation as in Eq. (2) in which more systems like B (i.e. B_1, B_2, \dots, B_N) are added.

Does one need to consider all the other quantum reference frames in the world to describe system B relative to a quantum reference frame A?

We are not sure that we understand the question of the Referee. If the meaning is that all degrees of freedom in the world should be taken into account, then this is not needed. For example, if the system C is an entire laboratory and A is a particle having only a few modes of motion (e.g. left or right), then changing to the reference frame of A, the laboratory would acquire only these modes of motion (left or right), despite being composed of a large number of subsystems.

v) At various places the authors claim that their approach is operational and does not use any external "absolute" reference frame. But, when in formula 2, they write observables x_A and p_B , these observables cannot be measured (defined) unless one has an ideal reference frame. In other words, using a (noisy) bounded quantum reference frame, one cannot measure observables x_B and p_b exactly. It seems that to define/measure these observables one needs an absolute ideal reference frame.

The Referee is correct in that we do not use any absolute degree of freedom, intended as an external reference frame, which is not described as a physical system from any point of view. Note that our approach is different to the bounded-size reference frames, in which the size of the reference frame itself limits the ability to measure the states of external systems. The observables we write in a quantum reference frame are relational, in that they do not rely on an external, absolute reference frame, but without the limitations present in the bounded-size reference frame approach. We clarified this by rephrasing the Introduction (p.3) and in the Appendix when comparing our and alternative approaches.

Please note that the changes from the initial version are marked out in blue in the revised version of the manuscript.

Point-by-point reply to Referee #2

We thank the Referee for their encouraging words. We are very happy to receive such positive feedback on our work. Please find below a point-by-point reply to the comments raised in the report.

6th page, 1st paragraph: At the beginning of the paragraph the authors state they consider C to be a unitary transformation, but in the text that follows they define C to be an isometry. Is the restriction to unitary transformations necessary? Said another way, where is the additional requirement that C is a unitary made use of?

The requirement that the transformation is a unitary is not fundamental for the transformation to be canonical. However, it is a condition that simplifies the treatment, for example in defining the scalar product and the adjoint action on the full density matrix. For this reason, we restrict to only unitary transformations. It would be interesting to go beyond this assumption, but this question is beyond the scope of the present work. Specifically, the unitarity requirement is used for the Hamiltonian in Eq. (5) to be hermitian, in the calculation of the transformed observables and variables in different quantum reference frames, and in the measurement section in the conservation of probabilities. In order to avoid confusion on this, we rephrased the corresponding paragraph on p. 6.

One could imagine a change between two quantum reference frames with the dimension of the Hilbert space associated with each reference frame is different.

In our manuscript, we only consider infinite-dimensional Hilbert spaces, which are all isomorphic. Therefore, there is always the same number of relational degrees of freedom, no matter in which reference frame we describe the external systems. However, it can happen that the Hilbert spaces are of different and finite dimensions. In this case, it might appear that there is loss of information (coherence) when moving from the Hilbert space of larger to the one of smaller dimension (please see also our answer to Referee #1, question 4.i). We added a comment on this on p. 8 of the revised version of the manuscript.

9th page, 4th paragraph: At first sight the Hamiltonian in Eq. (5) does not appear to be Hermitian: [equation] unless [equation]. However, in the preceding paragraph, the authors state that S is completely general and it is not clear whether this condition is indeed satisfied. Does the form of the operator given in Eq. (7) guarantee that the Hamiltonian relative to A is Hermitian? Does the fact that H must be Hermitian place a restriction on the functions $f_A(t)$ in Eq. (7)?

As mentioned in the previous point, the transformation S is indeed always unitary; therefore this problem does not apply to the cases considered. This is equivalent to saying that the operator f_A in Eq. (7) has to be hermitian. We rephrased the corresponding sentence in the manuscript to specify this (p.10 in the new version).

10th page, 2nd paragraph: In section III the authors identify symmetry transformations and note that these transformations imply the conservation of a dynamical observable. If possible, in the examples presented in sections IIIA and IIIB the authors should explicitly comment on the conservation of the dynamical variable associated with the symmetry transformations discussed.

We thank the Referee for this very interesting question. We decided to extend our work and add the Appendix “Conservation of the dynamical conserved quantities under quantum reference frame transformation” (Appendix D), in which we show that the set of dynamical conserved quantities in one reference frame can be mapped to the set of the dynamical conserved quantities in the new frame. In the specific case of a symmetric S transformation (i.e. it leaves the Hamiltonian in the same functional form), we show that it is possible to cast the transformed conserved quantities in the same functional form as the ones in the initial reference frame, but with all the labels A and C swapped. We then give some examples of how this is achieved in Section IIIA and IIIB.

Section IV: The authors propose a generalization of the weak equivalence principle for quantum reference frames. Can the authors imagine, even in principle, an experimental test of the weak equivalence principle for quantum reference frames? If so, this should be included in the manuscript.

We believe that the extension of the weak equivalence principle to quantum reference frames offers a good opportunity to test these ideas experimentally, and that it would be very interesting to do so. Existing techniques, such as in Refs. [24-26] of the revised version, which are able to measure relative degrees of freedom, would be good candidates to measure such effects. However, devising a concrete experimental proposal goes beyond the scope of this work. We added a short paragraph in Section IV, at the beginning of p. 18, and a comment in the Introduction (p. 4).

General comment: The authors cite in passing Changing quantum reference frames Palmer et al. Phys. Rev. A 89 042121 (2014), reference [7]. As the title suggests, Palmer et al. propose a framework for changing quantum reference frames –the same task the authors discuss in their manuscript. Note that Palmer et al. also emphasize the operational nature of their approach (and argue that to change quantum reference frames one must measure the relationship between the new and old reference frames). A fundamental difference between Palmer's et al. approach and the one presented in the author's manuscript is the appearance of a fundamental decoherence mechanism in the former. Seeing as both papers are discussing very similar (if not identical) tasks, both with operational considerations in mind, the differences and similarities between Palmer's et al. approach and the one presented in the author's manuscript should be discussed in detail.

We added the Appendix “Comparison between different approaches to quantum reference frames” (Appendix A) to the revised version of the manuscript, containing a detailed description of the relation between our approach and other existing approaches to quantum reference frames. In particular, there we compared in detail the approach of the paper by Palmer et al. with ours. We also added a new paragraph in the Introduction, which briefly explains the content of the paper “Changing quantum reference frames” by M. Palmer et al.

General comment: The authors conclude that superposition and entanglement are dependent on the quantum reference frame being employed. Presumably the notion of purity would also be dependent on the choice of quantum reference frame. The authors may wish to comment on this, although it is not necessary.

The Referee is correct in that the purity of the single system (say, B) is reference-frame dependent. However, since the transformation S is unitary, the purity of the joint state of A and B (or of B and C) is preserved under reference-frame transformation.

Reviewers' comments:

Reviewer #1 (Remarks to the Author):

First, I thank the authors for their response letter, which helped me to understand the result better. The revised version of the paper clarifies some of the points which were not clear before. However, in my opinion, there are still two remaining issues which are not addressed properly by the authors. Both of these issues have been discussed in my previous report, and also have been raised by referee 2.

Issue 1) It is still not clear how this approach can be extended to more general cases, beyond the simple examples considered in the paper. In my view, this is a fundamental issue, because it implies that we lack a clear understanding of how the change of reference frames should be defined according to this proposal.

Issue 2) It is still not clear that what are the essential differences between this approach and the previous approaches to Quantum Reference Frames?

In the revised version of the paper, the authors provide some responses to both of these questions. However, as I'll explain in the following, in my opinion their responses are not completely satisfactory/accurate.

In conclusion, I recommend the editors to give the authors the opportunity to clarify these points, before making the final decision on the manuscript.

Issue 1: How the definition can be extended beyond the simple cases considered in the paper?

In my previous report, I asked several questions trying to clarify this point. This helps to have a more clear understanding of the principle behind the authors' proposal. The authors' response has clarified some of these points, but some of the main questions remain unanswered.

i) In particular, in my previous report I asked:

"The transformation in Eq.(2) is defined only when A and C have the same Hilbert spaces. But, what if this is not the case? For instance, what happens if the new reference frame is a composite system? "

The authors address this question in the following paragraph in the revised manuscript:

"In this work, we assume that all Hilbert spaces are infinite-dimensional, and therefore isomorphic. However, when the Hilbert spaces are of finite but different dimensions, the transformation between the reference frames might lead to loss of information (i.e. coherence) for some degrees of freedom. The study of this case goes beyond the scope of the present work. "

First of all, strictly speaking, the claim about infinite-dimensional Hilbert spaces is not correct: Not all infinite-dimensional Hilbert spaces are isomorphic to each other (for example, a separable Hilbert space, i.e. a space with countable basis, is not isomorphic to a non-separable Hilbert space).

Secondly, and more importantly, isomorphism between two spaces is not sufficient to guarantee that the author's approach can be applied. Consider the following example: a particle moving on (i) a line, (ii) a 2-d flat surface, and (iii) a 2d sphere. In these examples, even though the Hilbert spaces are isomorphic, still the author's approach cannot be applied.

Another important example which shows this issue is the following: Suppose the reference frame is a particle with an internal (finite) spin degree of freedom. Again, regardless of the size of the spin degree of freedom, all such particles have isomorphic Hilbert spaces. Still, it is not clear how the author's approach can be applied in this case (This example is related to my previous question on composite systems).

I think the case of finite-dimensional systems is also fundamental, not only because of its relevance in quantum information, but also because it can potentially clarify the essence of this approach and how it is different from the previous approaches. So, I suggest the authors consider extending their work to include this case as well.

ii) In my previous report I asked about the scenarios where there are multiple reference frames. To clarify my point, consider the following example: Suppose there is a system B and multiple reference frames A_1, A_2, \dots, A_N .

The question is, given the description of the system B and the other reference frames with respect to reference frame A_1 , how should we find the description relative to another reference frame A_N ? One may directly apply the author's recipe to reference frames A_1 and A_N . But, alternatively, one can first apply it to go from reference frame A_1 to A_2 , and then go from A_2 to A_3 , etc. Aren't the final states different in these two cases? If so, how do the authors explain this?

iii) To explain how this approach works in general, it could be useful to give an explicit formula for a change of reference frame which corresponds to a general Galilean transformation.

Issue 2: Relation with previous works on quantum reference frames

In my previous report, I asked about the relation between the author's approach and the previous approaches to quantum reference frames (This question has also been asked by referee 2). The author's response is:

"Moreover, differently to other approaches in the literature, our work is not about the lack of a shared reference frame, and we don't consider our QRFs to be "bounded", feature which usually leads to an imprecise state assignment or to a noisy measurement result. In contrast, our QRFs allow to make state assignments to external systems with arbitrary precision."

I think this is not an accurate description of the previous works. For instance, the proposal based on twirling, discussed e.g. in Ref.4 and Ref 8 can be applied to both bounded and unbounded reference frames. Of course, the special case of bounded reference frames has been studied more extensively, because researchers have been interested to understand how the finite size of quantum reference frames limits the accuracy of measurements and how we can recover the classical limit. But this does not mean that the previous approaches cannot be applied to unbounded infinite-dimensional reference frames. For instance, the scenarios discussed in Fig. 3 in the manuscript can also be studied using the previous approaches (and I believe, one obtains similar conclusions).

In conclusion, I think the author still need to clarify what are the essential differences between their approach and the previous approaches to quantum reference frames. Specifically, a comparison with the approaches discussed in Ref 4 and Ref. 8 could be useful. These approaches can be applied to reference frames with arbitrary dimension (Note that Section II and IV of Ref 4 (Review of Modern Physics paper) are on the general problem of "Quantum Treatment of Reference Frames", and are not limited to bounded reference frames).

Reviewer #2 (Remarks to the Author):

The Author's have satisfactorily addressed my comments in their updated manuscript. In particular, they have added a thorough discussion of past work in an appendix, commenting specifically on the work of M. Palmer et al., which helps situate their work in the context of the existing literature on quantum reference frames.

I recommend that the updated manuscript be published in Nature Communications.

We thank the Referees for their comments. Following them, we have modified our paper accordingly and written a point-by-point reply to their concerns. All the modifications following the first round of comments are marked out in **BLUE** (same as in the previous version), and all the modifications following the second round of comments are marked out in **RED** in the manuscript.

For future reference, we state here our achievements in the manuscript (only mentioning what has never been done before):

- Extension of the usual reference frame transformation to superposition of positions, velocities, or accelerations using exclusively relational quantities;
- Introduction of a general method to build unambiguously the quantum reference frame transformation, once the type of transformation (i.e., translation, boost, etc.) has been chosen.
- Introduction of an extended notion of covariance for physical laws;
- Investigation of measurements of relational quantities in quantum reference frames, and of the unitarity of the dynamical evolution in quantum reference frames;
- Extension of the weak equivalence principle to a reference frame moving in a superposition of accelerations;
- Application of the formalism to a concrete situation (Section V), where the notion of quantum reference frame is crucial to define the rest frame of a quantum system.

Response to Referee #1

We thank the Referee for their comments. We have addressed them in the report and modified the manuscript accordingly (all the modifications are marked in red). It is our impression that in some of their questions the Referee is appealing to an external perspective, which has no meaning in our approach (please see below in the point-by-point reply). As we have stressed in our previous reply, one of the benefits of our formalism is that it does not require any external observer, and that everything is defined in terms of relational variables.

Reviewer #1 (Remarks to the Author):

First, I thank the authors for their response letter, which helped me to understand the result better. The revised version of the paper clarifies some of the points which were not clear before. However, in my opinion, there are still two remaining issues which are not addressed properly by the authors. Both of these issues have been discussed in my previous report, and also have been raised by referee 2.

Issue 1) It is still not clear how this approach can be extended to more general cases, beyond the simple examples considered in the paper. In my view, this is a fundamental issue, because it implies that we lack a clear understanding of how the change of reference frames should be defined according to this proposal.

Issue 2) It is still not clear that what are the essential differences between this approach and the previous approaches to Quantum Reference Frames?

In the revised version of the paper, the authors provide some responses to both of these questions. However, as I'll explain in the following, in my opinion their responses are not completely satisfactory/accurate.

In conclusion, I recommend the editors to give the authors the opportunity to clarify these points, before making the final decision on the manuscript.

Issue 1: How the definition can be extended beyond the simple cases considered in the paper?

In my previous report, I asked several questions trying to clarify this point. This helps to have a more clear understanding of the principle behind the authors' proposal. The authors' response has clarified some of these points, but some of the main questions remain unanswered.

In the manuscript, we do not only provide examples of the quantum reference frame transformation, but also a general method to quantize a reference frame. We are very precise as to how our transformation in Eq. (2) can be generalised. In fact, Eq. (7) is the most general transformation which can be achieved in the 1-dimensional case, when A and B do not interact. The general procedure to follow is: identify the transformation to the new reference frame classically, quantize it by using the equations of motion of the new reference frame. This constitutes the central part of Eq. (7). We then choose the generalized parity and swap operator so that the equations of motion of A are the opposite of the equations of motion of C, i.e., they are equal up to a minus sign. We have added a paragraph on p. 12 of the manuscript to explain this general procedure.

We further apply our method to four relevant cases, which we believe are highly nontrivial: the time independent transformation of Eq. (2), the superposition of translations, the superposition of Galilean boosts, the extension of the weak equivalence principle. These transformations show, for the first time in the literature, a way to "jump" to a reference frame in a superposition of positions, velocities, or accelerations.

i) In particular, in my previous report I asked:

"The transformation in Eq.(2) is defined only when A and C have the same Hilbert spaces. But, what if this is not the case? For instance, what happens if the new reference frame is a composite system? "

The authors address this question in the following paragraph in the revised manuscript:

"In this work, we assume that all Hilbert spaces are infinite-dimensional, and therefore isomorphic. However, when the Hilbert spaces are of finite but different dimensions, the transformation between the reference frames might lead to loss of information (i.e. coherence) for some degrees of freedom. The study of this case goes beyond the scope of the present work. "

First of all, strictly speaking, the claim about infinite-dimensional Hilbert spaces is not correct: Not all infinite-dimensional Hilbert spaces are isomorphic to each other (for example, a separable Hilbert space, i.e. a space with countable basis, is not isomorphic to a non-separable Hilbert space).

The Referee is correct on the claim about infinite-dimensional Hilbert spaces. Strictly speaking, we work with a Hilbert space isomorphic to $L^2(\mathbb{R})$ with standard integration measure dx . We added this in the main text on p. 8. When the new reference frame is a composite system as seen from the old reference frame, our formalism can be straightforwardly applied, using a subset of the degrees of freedom (for instance, the centre of mass degrees of freedom, or the degrees of freedom of one of the particle if the composite system contains several particles) to define the relational quantities. The remaining degrees of

freedom can simply be treated as an external system. In order to clarify this point, we have added an example in Section II, p. 8.

Secondly, and more importantly, isomorphism between two spaces is not sufficient to guarantee that the author's approach can be applied. Consider the following example: a particle moving on (i) a line, (ii) a 2-d flat surface, and (iii) a 2d sphere. In these examples, even though the Hilbert spaces are isomorphic, still the author's approach cannot be applied.

We disagree with the Referee on this point. The first important observation is that in our relational formalism, a system moves on a line, on a 2-sphere, or along a plane *from the point of view* of the initial reference frame. This is an instance of a question where it seems that the Referee adopts an external perspective. Having said that, we show in the following how our formalism can be applied to this case. Whenever a system moves in less than 3 dimensions (the dimension of the space where we live), there is a physical reason for that, such as an interaction between the system and another physical system (e.g., a guide. Please note that the effect of the interaction can be such that in the remaining degree of freedom the Hamiltonian is a 1D free evolution), constraining the former system to move in effectively fewer dimensions. Therefore, the procedure to apply in this case is: embed the system in a 3 dimensional space; consider the interaction and the degrees of freedom of the guide. The system composed of the initial system plus the guide then has the correct number of degrees of freedom (after imposing the constraint), and the transformation of reference frame can be straightforwardly applied (please, see also our reply to the next point).

Another important example which shows this issue is the following: Suppose the reference frame is a particle with an internal (finite) spin degree of freedom. Again, regardless of the size of the spin degree of freedom, all such particles have isomorphic Hilbert spaces. Still, it is not clear how the author's approach can be applied in this case (This example is related to my previous question on composite systems).

I think the case of finite-dimensional systems is also fundamental, not only because of its relevance in quantum information, but also because it can potentially clarify the essence of this approach and how it is different from the previous approaches. So, I suggest the authors consider extending their work to include this case as well.

We strongly disagree with the Referee that omitting the finite-dimensional case could in any case diminish our current results. On the contrary, our work is novel in so many different ways (please, see the list above) that its main message - the possibility of extending the covariance of physical laws to quantum reference frames - might remain hidden by adding another technical discussion that does not contribute to the main message of the paper. We believe that the question of the Referee is interesting, but it would deserve a study of its own, and a dedicated publication. In this paper we consider, in general, the case of spinless particles, and also the spin degrees of freedom, in the specific case of Section V, where the spin of particle A is treated as another external system. In this specific example, the quantum reference frame transformation is key to define the rest frame of a particle.

Preliminary considerations indicate that the spin transformation can be treated analogously to the one-dimensional formalism we describe in the present manuscript, introducing the transformation corresponding to the "superposition of rotations". We here notice that, **by construction**, our formalism cannot result in a transformation between two non-isomorphic Hilbert spaces. The reason is

that we do not choose the two Hilbert spaces independently (which would be an external viewpoint). The second Hilbert space is **constructed** from the first one from the relational degrees of freedom. Hence, it is by definition isomorphic to the first Hilbert space, i.e., each state in the initial Hilbert space will be transformed by our transformation into a state of the second Hilbert space. As an example, if A is a spin system from the point of view of C (i.e., a system with two possible outcomes of a measurement), C will effectively be a spin system from the point of view of A.

ii) In my previous report I asked about the scenarios where there are multiple reference frames. To clarify my point, consider the following example: Suppose there is a system B and multiple reference frames A_1, A_2, \dots, A_N .

The question is, given the description of the system B and the other reference frames with respect to reference frame A_1 , how should we find the description relative to another reference frame A_N ? One may directly apply the author's recipe to reference frames A_1 and A_N . But, alternatively, one can first apply it to go from reference frame A_1 to A_2 , and then go from A_2 to A_3 , etc. Aren't the final states different in these two cases? If so, how do the authors explain this?

The Referee has already asked this question in their previous report. To answer this question, we have added in the previous round of comments the Appendix C. There, we showed with an explicit calculation that the transitive property holds, i.e., going from the reference frame C to A yields the same result as going from C to B, and then from B to A. This argument can be straightforwardly re-adapted to the case of N systems as asked by the Referee, by composing different transformations. For clarity, we have added this case to Appendix C, showing explicitly that the transformation from A_0 to A_N has the same effect as transforming from A_0 to A_1 , then to A_2 , and finally to A_N . Therefore, our formalism is not affected by the problem mentioned by the Referee.

iii) To explain how this approach works in general, it could be useful to give an explicit formula for a change of reference frame which corresponds to a general Galilean transformation.

Eq. (7) provides the most general expression for a Galilean transformation (in fact, it can be even more general than a Galilean transformation) in a one-dimensional system. For clarity, we have explicitly stated that the (extended) Galilean transformations in one dimension are a particular case of Eq. (7) on p. 12 of the manuscript. For what concerns the generalisation to more than one dimension, it amounts to consider the transformation corresponding to the "superposition of rotations". We believe that an explicit derivation of this transformation is technically cumbersome, but does not bring any new insights to the conceptual understanding of how the physical description of systems results from a QRF and how these descriptions relate when changing QRFs. The paper already contains a very significant amount of new material (please, refer to the list at the beginning of our reply) and we see a danger that extending its content to new results may in fact be counterproductive and screen its main message.

Issue 2: Relation with previous works on quantum reference frames

In my previous report, I asked about the relation between the author's approach and the previous approaches to quantum reference frames (This question has also been asked by referee 2). The author's response is:

"Moreover, differently to other approaches in the literature, our work is not about the lack of a shared reference frame, and we don't consider our QRFs to be "bounded", feature which usually leads to an imprecise state assignment or to a noisy measurement result. In contrast, our QRFs allow to make state assignments to external systems with arbitrary precision."

I think this is not an accurate description of the previous works. For instance, the proposal based on twirling, discussed e.g. in Ref.4 and Ref 8 can be applied to both bounded and unbounded reference frames. Of course, the special case of bounded reference frames has been studied more extensively, because researchers have been interested to understand how the finite size of quantum reference frames limits the accuracy of measurements and how we can recover the classical limit. But this does not mean that the previous approaches cannot be applied to unbounded infinite-dimensional reference frames. For instance, the scenarios discussed in Fig. 3 in the manuscript can also be studied using the previous approaches (and I believe, one obtains similar conclusions).

In conclusion, I think the author still need to clarify what are the essential differences between their approach and the previous approaches to quantum reference frames. Specifically, a comparison with the approaches discussed in Ref 4 and Ref. 8 could be useful. These approaches can be applied to reference frames with arbitrary dimension (Note that Section II and IV of Ref 4 (Review of Modern Physics paper) are on the general problem of "Quantum Treatment of Reference Frames", and are not limited to bounded reference frames).

The Referee is incomplete in reproducing our reply. Following the previous round of comments, we have added a whole new Appendix A to discuss the relation between the previous approaches to QRFs and ours. Referee 2 writes that "The Author's have satisfactorily addressed my comments in their updated manuscript. In particular, they have added a thorough discussion of past work in an appendix, commenting specifically on the work of M. Palmer et al., which helps situate their work in the context of the existing literature on quantum reference frames."

We agree with the Referee that previous proposals can be applied to unbounded reference frames. Our chosen wording was motivated by the fact that bounded reference frames have been studied more extensively in the literature, fact which is acknowledged by the Referee. This is why we wrote, after the Referee's first suggestion to compare our result to the existing literature, that "The literature [4–12] **mainly** focusses on a) the consequences of the lack of a shared reference frame for quantum information tasks, on b) the generalisation of the fact that superselection rules can be overcome by choosing an appropriate quantum system as a reference frame, and on c) "bounded" reference frames" at the beginning of the third paragraph in Appendix A.

We have discussed in our previous reply in great detail how our approach compares to Ref. 8 (please see Appendix A), dedicating over half a page to this comparison.

As we have already written in Appendix A in the previous round of comments, a fundamental difference between our proposal and others is that all the other proposals discuss how to find relational quantities starting from an absolute, external reference frame. We do not need to apply these methods (like the "twirling") to address this question, because our formalism is genuinely relational from the very start. We have added a more explicit explanation of this point in Appendix A and in the Introduction (p. 3). Our focus is, given the relational quantities in different QRF, how they transform, and which physical laws are invariant under these transformations.

The only paper, to the best of our knowledge, considering the transformation between two quantum reference frames, which can therefore be compared to our approach, is Ref. [8]. We have already highlighted in Appendix A how, for instance, the POVM involved in the procedure of changing the reference frame makes this approach substantially different to ours.

Finally, none of the approaches to quantum reference frames discussed in the literature deals with an extension of the symmetry transformation and an extended notion of covariance of physical laws deriving from considering a quantum reference frame. As is reflected in the title of our paper, this is the main, novel message that we would like to convey. We have added a paragraph to stress this point at the end of Appendix A.

Response to Referee #2

We thank the Referee for their encouraging words and for their helpful feedback. We are very happy that they recommend our paper for publication.

REVIEWERS' COMMENTS:

Reviewer #1 (Remarks to the Author):

I thank the authors for their response.

In my previous report I had two main concerns and questions: First, I asked the authors to explain how their results can be generalized to the cases where the Hilbert spaces of quantum reference frames are not isomorphic, including the case of systems with finite Hilbert spaces. The authors responded:

"We strongly disagree with the Referee that omitting the finite-dimensional case could in any case diminish our current results. On the contrary, our work is novel in so many different ways (please, see the list above) that its main message – the possibility of extending the covariance of physical laws to quantum reference frames - might remain hidden by adding another technical discussion that does not contribute to the main message of the paper. We believe that the question of the Referee is interesting, but it would deserve a study of its own, and a dedicated publication. "

Although I do not agree with the authors that the case of finite-dimensional Hilbert spaces is just "another technical discussion", nevertheless, I accept the authors' argument that this case can be left for future works. Having said this, I strongly encourage the authors to add this case to the paper, if they can solve it.

My second main request from the authors was to clarify the relation between this approach and the previous works on quantum reference frames. This was the second time I was asking the authors to explain this. In response, the authors say:

"The only paper, to the best of our knowledge, considering the transformation between two quantum reference frames, which can therefore be compared to our approach, is Ref. [8]."

And

"Finally, none of the approaches to quantum reference frames discussed in the literature deals with an extension of the symmetry transformation and an extended notion of covariance of physical laws deriving from considering a quantum reference frame. As is reflected in the title of our paper, this is the main, novel message that we would like to convey. We have added a paragraph to stress this point at the end of Appendix A."

Unfortunately, both of these claims are wrong. This topic has been studied for a long time, and there are many other papers on transformations between quantum reference frames. As I have mentioned before, the authors can simply check the Review of Modern Physics paper on quantum reference frames (Ref 4), and the references therein.

An important example is a well-known paper by Aharonov and Kaufherr from 1984, titled "Quantum frames of reference" (Ref 3 of the submitted paper, Phys.Rev.D 30.2, 368). In this influential paper, Aharonov and Kaufherr study transformations between quantum reference frames. Moreover, in the abstract of their paper Aharonov and Kaufherr say:

"The main result of the present work is a formalism wherein the principle of equivalence is

extended to reference frames described by quantum states.”

But these are exactly the problems studied in the submitted manuscript: It first studies transformations between two quantum reference frames, and then presents a formulation of equivalence principle for quantum reference frames. So, a natural question is how the submitted manuscript relates to the well-known work of Aharonov and Kaufherr? Are these two approaches identical, or the authors believe Aharonov-Kaufherr approach is different/wrong?

Unfortunately, the authors do not address this natural question in the paper or in their response letter. The manuscript only briefly mentions Aharonov-Kaufherr paper in the introduction, where it says “In Ref. [3] it was shown that it is possible to consistently formulate quantum theory without appealing to classical reference frames as well-localized laboratories of infinite mass.”

To understand the relation between the two papers, I checked Aharonov-Kaufherr paper more carefully. As far as I understand, the two approaches are, at least, closely related, if not identical. In particular, Equation (24) of Aharonov-Kaufherr paper looks essentially identical to Equation (2) of the submitted manuscript, which defines the transformations between quantum reference frames. It seems that the only difference is that Aharonov and Kaufherr first start with the state of quantum reference frames relative to a classical reference frame, and then use this to find the unitary that transforms state from one quantum reference frame to another. Their final result is independent of the classical reference frame though (they remove one degree of freedom by assuming the total momentum is fixed, without loss of generality).

In summary, I cannot recommend publication of the submitted manuscript in its present form. Despite my previous requests, the manuscript still does not discuss the related previous works and contributions properly. An important example is the well-known paper by Aharonov- Kaufherr from 1984, which studies exactly the same problems studied in this manuscript, and claims to establish exactly the same results. In fact, I believe the Aharonov-Kaufherr results and the main results of the submitted manuscript are, at least, closely related, if not identical. Therefore, the paper also does not satisfy the novelty criterion required for publication in Nature Communications.

Reviewer #2 (Remarks to the Author):

The following remarks are in response to the concerns raised by Referee #1.

1) How can the work be extended beyond the cases considered?

I do believe Giacomini et al. have already gone far beyond what has been done in the past. In particular, they introduced a formalism that can handle reference frames associated with noncompact groups (e.g. the Galilean group), a topic not discussed in the Review of Modern Physics article (Ref. 4). This in itself is nontrivial, and something I believe the recent literature has not focused on. The authors also state that their method is applicable within special and general relativistic contexts, pointing to another possible case in which the formalism can be applied.

2) What are the essential differences between this approach and the previous approaches to quantum reference frames?

With regard to previous work, referee #1 is correct to point out the similarities between Eq. (24) of the Aharonov and Kaufherr and Eq. (2) of the submitted manuscript. I hadn't appreciated this similarity before, and the authors should at the very least have an opportunity to address this point directly.

Nonetheless, Giacomini et al. have gone significantly beyond what was done by Aharonov and Kaufherr. In particular, they motivate and give a different interpretation of the unitary operator that changes reference frames, which makes no reference to a classical reference frame. This is an important conceptual point, as most (if not all) past works on quantum reference frames begin with some notion of a classical reference frame.

Perhaps most importantly, Giacomini et al. demonstrate explicitly how entanglement and superposition are related under changes of a quantum reference frame. This point is surely to be of foundational importance, and something not at all discussed in the work of Aharonov and Kaufherr. For me, this is the really novel contribution from Giacomini et al.

In addition, Giacomini et al. also present a generalization of the “weak equivalence principle” applicable to quantum reference frames, which is distinct from the “principle of equivalence” discussed by Aharonov and Kaufherr. Furthermore, Giacomini et al. give a quantum notion of a rest frame and discuss the measurement procedure as seen from different reference frames. These contributions add to the importance of the authors’ manuscript.

In sum, the submitted manuscript is a significant contribution within the foundations community and exceeds the novelty criterion for publication in Nature Communications. I still recommend publication of this manuscript.

Reviewer #3 (Remarks to the Author):

I have read the newly revised version of the paper and the authors’ responses to the Referees 1 and 2 in the second round.

First off, I am highly impressed by this paper, it really does break important new ground. While Referee 2 recommends publication, Referee 1 asked for some further clarifications. I find the authors’ replies to Referee 1 convincing and I find that the corresponding revisions that they implemented fully address these points.

In particular, Referee 1 asked for further clarifications regarding the generalizability of the new methods. While there is always more that could be done, I agree with the authors’ reply.

In particular, I agree with the authors that, while adding a study of the finite-dimensional case with possibly unequal dimensions would be an interesting and technically nontrivial addition, the case of infinite-dimensional (and separable) Hilbert spaces that the authors are considering here is itself of very high interest since this case already covers a huge number of physical systems.

Referee 1 also raises the question as to how the case of dimensionally restricted motion would be covered. The authors explain, convincingly, that in any such case, the dimensional restriction must be enforced by some physical system (they give the example of a guide), which then enables one to use their full-dimensional techniques.

Further, Referee 1 raises the question as to the transitivity of frame changes for more than two frames. The authors show that the transitivity does hold. Referee 1 also asks for the most general Galileian transformation. The authors provide it in Eq.(7), in one dimension and explain how it generalizes.

Referee 1 also asked for more discussion of the relationship of the present work to prior work on quantum reference frames. I think that the authors have added sufficient material on this by adding Appendix A. They clarify, in particular, the novelty of the background independence of the

new approach (in the sense of no need for an external observer), which I think is indeed an major advance since, in particular, it leads to a new quantum generalization of the notions of covariance and the equivalence principle.

In conclusion, in my opinion, the paper makes truly major contributions and it is technically sound. I highly recommend publication.

We thank the Referees for their comments. We have replied to the points raised below.

Reviewer #1 (Remarks to the Author):

We thank the Referee for their comments. Please find below a point-by-point reply.

I thank the authors for their response.

In my previous report I had two main concerns and questions: First, I asked the authors to explain how their results can be generalized to the cases where the Hilbert spaces of quantum reference frames are not isomorphic, including the case of systems with finite Hilbert spaces. The authors responded:

“We strongly disagree with the Referee that omitting the finite-dimensional case could in any case diminish our current results. On the contrary, our work is novel in so many different ways (please, see the list above) that its main message – the possibility of extending the covariance of physical laws to quantum reference frames - might remain hidden by adding another technical discussion that does not contribute to the main message of the paper. We believe that the question of the Referee is interesting, but it would deserve a study of its own, and a dedicated publication. “

Although I do not agree with the authors that the case of finite-dimensional Hilbert spaces is just "another technical discussion", nevertheless, I accept the authors' argument that this case can be left for future works. Having said this, I strongly encourage the authors to add this case to the paper, if they can solve it.

Despite the fact that we believe that the case of finite dimensions is an interesting generalisation of our formalism, we have decided not to add it to the present manuscript, because we believe it deserves being address in a dedicated work. We are happy that the Referee accepts our argument.

My second main request from the authors was to clarify the relation between this approach and the previous works on quantum reference frames. This was the second time I was asking the authors to explain this. In response, the authors say:

“The only paper, to the best of our knowledge, considering the transformation between two quantum reference frames, which can therefore be compared to our approach, is Ref. [8].”

And

“Finally, none of the approaches to quantum reference frames discussed in the literature deals with an extension of the symmetry transformation and an extended notion of covariance of physical laws deriving from considering a quantum reference frame. As is reflected in the title of our paper, this is the main, novel message that we would like to convey. We have added a paragraph to stress this point at the end of Appendix A.”

Unfortunately, both of these claims are wrong. This topic has been studied for a long time, and there are many other papers on transformations between quantum reference frames. As I have mentioned before, the authors can simply check the Review of Modern Physics paper on quantum reference frames (Ref 4), and the references therein.

We disagree with the Referee, and stand by our previous reply. The Referee did not provide any new, concrete evidence for their claims, but only cited the Rev. Mod. Phys (Ref. [4]). We believe that we have addressed the content of that review extensively in our previous replies and in the Appendix. We can not comment further, as there is no concrete indication of where our analysis is wrong, incomplete, or lacks novelty.

An important example is a well-known paper by Aharonov and Kaufherr from 1984, titled “Quantum frames of reference” (Ref 3 of the submitted paper, Phys.Rev.D 30.2, 368). In this influential paper,

Aharonov and Kaufherr study transformations between quantum reference frames. Moreover, in the abstract of their paper Aharonov and Kaufherr say:

"The main result of the present work is a formalism wherein the principle of equivalence is extended to reference frames described by quantum states."

But these are exactly the problems studied in the submitted manuscript: It first studies transformations between two quantum reference frames, and then presents a formulation of equivalence principle for quantum reference frames. So, a natural question is how the submitted manuscript relates to the well-known work of Aharonov and Kaufherr? Are these two approaches identical, or the authors believe Aharonov-Kaufherr approach is different/wrong?

Unfortunately, the authors do not address this natural question in the paper or in their response letter. The manuscript only briefly mentions Aharonov-Kaufherr paper in the introduction, where it says "In Ref. [3] it was shown that it is possible to consistently formulate quantum theory without appealing to classical reference frames as well-localized laboratories of infinite mass."

To understand the relation between the two papers, I checked Aharonov-Kaufherr paper more carefully. As far as I understand, the two approaches are, at least, closely related, if not identical. In particular, Equation (24) of Aharonov-Kaufherr paper looks essentially identical to Equation (2) of the submitted manuscript, which defines the transformations between quantum reference frames. It seems that the only difference is that Aharonov and Kaufherr first start with the state of quantum reference frames relative to a classical reference frame, and then use this to find the unitary that transforms state from one quantum reference frame to another. Their final result is independent of the classical reference frame though (they remove one degree of freedom by assuming the total momentum is fixed, without loss of generality).

The Referee is correct that Eq. (24) of the paper by Aharonov and Kaufherr is indeed similar to one of the transformations we find in our paper. This is not surprising, because it is the way to achieve a transformation to relative coordinates. However, there is a fundamental difference, which does not only reside in the fact that the authors rely on the presence of an absolute reference frame, but also in the fact that they assign a position operator to the reference frame in which physics is described. In our formalism, because of its fully relational flavour, this would not have any meaning. In addition to this, we find a much more general class of transformations than the one shown in the paper by Aharonov and Kaufherr (in fact, we provide a general method to quantize a reference frame). Finally, it is not clear whether the principle of equivalence in the work by Aharonov and Kaufherr is related to our notion of extended covariance. In particular, the authors assume the principle of equivalence (intended, to the best of our understanding, as the equivalence of all reference frames), while we derive it from the quantum reference frame transformation. In addition, in our extension of the weak equivalence principle we discuss the equivalence between physics in an accelerated reference frame and physics in a uniform gravitational field. As a result of all these considerations, we firmly believe that our paper has many novel results and aspects which are not covered in the literature.

In summary, I cannot recommend publication of the submitted manuscript in its present form. Despite my previous requests, the manuscript still does not discuss the related previous works and contributions properly. An important example is the well-known paper by Aharonov-Kaufherr from 1984, which studies exactly the same problems studied in this manuscript, and claims to establish exactly the same results. In fact, I believe the Aharonov-Kaufherr results and the main results of the submitted manuscript are, at least, closely related, if not identical. Therefore, the paper also does not satisfy the novelty criterion required for publication in Nature Communications.

We disagree with the conclusions of the Referee for the reasons listed above.

Reviewer #2 (Remarks to the Author):

We thank the Referee for their positive feedback. We agree with all the arguments of the Referee, and are very happy that they recommend our paper for publication.

The following remarks are in response to the concerns raised by Referee #1.

1) How can the work be extended beyond the cases considered?

I do believe Giacomini et al. have already gone far beyond what has been done in the past. In particular, they introduced a formalism that can handle reference frames associated with noncompact groups (e.g. the Galilean group), a topic not discussed in the Review of Modern Physics article (Ref. 4). This in itself is nontrivial, and something I believe the recent literature has not focused on. The authors also state that their method is applicable within special and general relativistic contexts, pointing to another possible case in which the formalism can be applied.

2) What are the essential differences between this approach and the previous approaches to quantum reference frames?

With regard to previous work, referee #1 is correct to point out the similarities between Eq. (24) of the Aharonov and Kaufherr and Eq. (2) of the submitted manuscript. I hadn't appreciated this similarity before, and the authors should at the very least have an opportunity to address this point directly.

Please, see the reply to Referee 1 for a comment on this point.

Nonetheless, Giacomini et al. have gone significantly beyond what was done by Aharonov and Kaufherr. In particular, they motivate and give a different interpretation of the unitary operator that changes reference frames, which makes no reference to a classical reference frame. This is an important conceptual point, as most (if not all) past works on quantum reference frames begin with some notion of a classical reference frame.

Perhaps most importantly, Giacomini et al. demonstrate explicitly how entanglement and superposition are related under changes of a quantum reference frame. This point is surely to be of foundational importance, and something not at all discussed in the work of Aharonov and Kaufherr. For me, this is the really novel contribution from Giacomini et al.

In addition, Giacomini et al. also present a generalization of the "weak equivalence principle" applicable to quantum reference frames, which is distinct from the "principle of equivalence" discussed by Aharonov and Kaufherr. Furthermore, Giacomini et al. give a quantum notion of a rest frame and discuss the measurement procedure as seen from different reference frames. These contributions add to the importance of the authors' manuscript.

In sum, the submitted manuscript is a significant contribution within the foundations community and exceeds the novelty criterion for publication in Nature Communications. I still recommend publication of this manuscript.

Reviewer #3 (Remarks to the Author):

We thank the Referee for their positive feedback and encouraging words on our work. We are very happy that they find that our paper makes major contributions and highly recommend its publication.

I have read the newly revised version of the paper and the authors' responses to the Referees 1 and 2 in the second round.

First off, I am highly impressed by this paper, it really does break important new ground. While Referee 2 recommends publication, Referee 1 asked for some further clarifications. I find the authors' replies to Referee 1 convincing and I find that the corresponding revisions that they implemented fully address these points.

In particular, Referee 1 asked for further clarifications regarding the generalizability of the new methods. While there is always more that could be done, I agree with the authors' reply.

In particular, I agree with the authors that, while adding a study of the finite-dimensional case with possibly unequal dimensions would be an interesting and technically nontrivial addition, the case of infinite-dimensional (and separable) Hilbert spaces that the authors are considering here is itself of very high interest since this case already covers a huge number of physical systems.

Referee 1 also raises the question as to how the case of dimensionally restricted motion would be covered. The authors explain, convincingly, that in any such case, the dimensional restriction must be enforced by some physical system (they give the example of a guide), which then enables one to use their full-dimensional techniques.

Further, Referee 1 raises the question as to the transitivity of frame changes for more than two frames. The authors show that the transitivity does hold. Referee 1 also asks for the most general Galileian transformation. The authors provide it in Eq.(7), in one dimension and explain how it generalizes.

Referee 1 also asked for more discussion of the relationship of the present work to prior work on quantum reference frames. I think that the authors have added sufficient material on this by adding Appendix A. They clarify, in particular, the novelty of the background independence of the new approach (in the sense of no need for an external observer), which I think is indeed an major advance since, in particular, it leads to a new quantum generalization of the notions of covariance and the equivalence principle.

In conclusion, in my opinion, the paper makes truly major contributions and it is technically sound. I highly recommend publication.